



# A global eddying hindcast ocean simulation with OFES2

Hideharu Sasaki[1], Shinichiro Kida[2,1], Ryo Furue[1], Hidenori Aiki[3,1], Nobumasa Komori[1], Yukio Masumoto[4,1], Toru Miyama[1], Masami Nonaka[1], Yoshikazu Sasai[5], Bunmei Taguchi[6,1]

[1]Application Laboratory, Japan Agency for Marine-Earth Science and Technology, Yokohama, Japan
[2]Research Institute for Applied Mechanics, Kyushu University, Kasuga, Japan
[3]Institute for Space-Earth Environmental Research, Nagoya University, Nagoya, Japan
[4]Graduate School of Science, The University of Tokyo, Tokyo, Japan
[5]Research Institute for Global Change, Japan Agency for Marine-Earth Science and Technology, Yokosuka, Japan
[6]Faculty of Sustainable Design, University of Toyama, Toyama, Japan

*Correspondence to*: Hideharu Sasaki (sasaki@jamstec.co.jp)

**Abstract.** A quasi-global eddying ocean hindcast simulation using a new version of our model called "OFES2" (Ocean General Circulation Model for the Earth Simulator version 2) was conducted to overcome several issues with unrealistic properties in its previous version "OFES". This paper describes the model and the simulated oceanic fields in OFES2 compared with OFES and also observed data. A sea-ice model and a tidal mixing scheme were implemented in OFES2, which was forced by a newly created surface atmospheric dataset called JRA55-do and simulated the oceanic fields from 1958 to 2016. We found several improvements in OFES2 over OFES: smaller biases in the global sea surface temperature and sea surface salinity and the water properties in the Indonesian and Arabian Seas. The time series of the Niño3.4 and Indian Ocean Dipole (IOD) indexes are somewhat better in OFES2 than in OFES. Unlike the previous version, OFES2 reproduces more realistic anomalous low sea surface temperatures during a positive IOD event. One possible cause for these improvements in El Niño and IOD events is the replacement of the atmospheric dataset. On the other hand, several issues remained unrealistic, such as the pathways of the Kuroshio and Gulf Stream and the unrealistic spreading of salty Mediterranean overflow. Given the worldwide use of the previous version and the improvements presented here on it, the output from OFES2 will be useful in studying various oceanic phenomena with broad spatiotemporal scales.

## 1 Introduction

The global ocean includes phenomena with various spatial scales. Basin-scale circulations occur over thousands of kilometers, while oceanic fronts, western boundary currents, and the Antarctic Circumpolar Current (ACC) have widths of approximately or less than 100 km. Mesoscale eddies, ubiquitous around these currents and in the ocean interior, have a spatial scale of a few tens of kilometers in the sub-Arctic ocean to a few hundred kilometers in the subtropics (Chelton et al., 1998). The location and strength of oceanic fronts, currents, and mesoscale eddies also change over time (e.g., Sasaki and Schneider, 2011; Qiu and Chen, 2010; Zhai et al., 2008).





Observations are crucial for understanding the ocean, but their data coverage and resolution are limited. Since the 2000s, gridded hydrographic products based on Argo float observations (e.g., Roemmich et al., 2009, Hosoda et al., 2008) have been able to capture global ocean properties at a resolution of approximately 300 km. However, such a spatial resolution is
not adequate to observe currents, mesoscale eddies, and frontal structures. Satellite observations can provide high-resolution data of the sea surface height (SSH) and temperature (SST), etc. but are limited to surface measurements. Global eddying simulations, therefore, have become a useful and convenient tool for understanding the ocean. Computational power has increased exponentially, and over past decades, several research groups have been conducting global eddying ocean simulations at horizontal resolutions of approximately 10 km using the Parallel Ocean Program (POP, Maltrud and McClean,
2005), the Hybrid Coordinate Ocean Model (HYCOM, Chassignet et al., 2006), the Max Planck Institute ocean model (MPIOM, Jungclaus et al., 2013), and the Ocean General Circulation Model (OGCM) for the Earth Simulator (OFES, Masumoto et al., 2004).

Realistic long-term hindcast global eddying ocean simulation outputs have been widely used in the community and have provided unprecedented information about oceanic phenomena on wide spatiotemporal scales in areas where observational
data are limited. These simulations create a significant amount of data, which are very informative because the data exhibit oceanic phenomena from around the globe with the scales from mesoscales to large scales and their variations from intraseasonal to decadal timescales. Sharing simulation outputs among the community is crucial, and such use of OFES (Sasaki et al., 2008) has led to research achievements in various topics (see details in Masumoto 2010), such as in oceanic phenomena from intraseasonal (e.g., Hu et al. 2018) to decadal variations (e.g., Taguchi et al. 2017) and mesoscale eddies
(e.g., Aoki et al. 2016). However, numerical models are not perfect. Model deficiencies and biases exist, and the usage of simulation outputs in the community has led to findings of where these limitations exist and their possible causes. One of the major problems of OFES seems to be its surface wind stress fields. Kutsuwada et al. (2019) showed that the thermocline depth in the subtropical northwestern Pacific was too shallow due to unrealistic wind stress. The second problem is the lack of tidally induced vertical mixing. Masumoto et al. (2008) found unrealistic water properties within the Indonesian seas,
where tidally induced vertical mixing is considered significant (Ffield and Gordon, 1996). The third problem is the lack of sea ice. Therefore, the sea surface salinity in OFES was strongly restored to monthly climatological observations.

The goal of this paper is to present improved hindcast simulation outputs from OFES. This model was forced by surface forcing based on 3-hourly atmospheric reanalysis data at a finer horizontal resolution. A tidal mixing scheme and a sea-ice model were added, and we call the standard hindcast simulation using this new version OFES2 (Fig. 1). In this paper, we
will present how model outputs in OFES2 have improved from OFES. Section 2 describes OFES2, and Section 3 examines its simulated mean oceanic fields. Section 4 examines the time variability based on climate indexes of El Niño and the Indian Ocean Dipole Mode (IOD). We will further examine the IOD events and highlight the simulated SST distribution around the eastern pole of the IOD. A summary and discussion are provided in Section 5.



## 2 Descriptions of OFES2 compared with OFES

OFES2 is an update of a quasi-global eddying hindcast simulation: OFES (Sasaki et al., 2008). It is based on Modular Ocean Model (MOM) version 3 (Pacanowski and Griffies, 1999) and utilizes the latitude and longitude grid system. The horizontal resolution of 0.1° remains the same as that in OFES, but the model setup and parameterization are added to reduce the model biases that exist in OFES. The model configuration of OFES2 will be described first, and the differences from OFES will be described next.

The domain extends from 76° S to 76° N without polar regions. The horizontal resolution is 0.1°, and the number of vertical levels is 105 with a maximum depth of 7,500 m. The thickness of the first layer is 5 m, and 55 levels exist within the upper 500 m. We constructed the bottom topography with partial bottom cells (Adcroft et al., 1997) using the bathymetry dataset ETOPO1 (Amante and Eakins, 2009). Although the model domain does not include the polar region, a sea-ice model (Komori et al., 2005) is used to simulate the Antarctic Sea and the Subarctic seas, including the Sea of Okhotsk, more

realistically. The sea-ice model employs two-category, zero-layer thermodynamics (Hibler 1979) and elastic-viscous-plastic rheology (Hunke and Dukowicz, 2002).

    A biharmonic operator is used for horizontal mixing to suppress computational noise with a viscosity of $-27 \times 10^9$ m$^4$ s$^{-1}$ and a diffusivity of $-9 \times 10^9$ m$^4$ s$^{-1}$. The drag coefficient is $2.5 \times 10^{-3}$ (non-dimensional) for linear bottom drag. For vertical mixing, we added diffusivities from the tidal mixing scheme developed by Jayne and St. Laurent (2001) and St. Laurent et al.

(2002) to those estimated from the mixed layer vertical mixing scheme (Noh and Kim, 1999). In the tidal mixing scheme, the three-dimensional diffusivities are estimated from the energy flux at the ocean bottom and the local buoyancy frequency with the parameters of dissipation efficiency, mixing efficiency, and vertical scale. These parameters are the same as those used by St. Laurent et al. (2002). We used a constant barotropic tidal current of K1 and M2 tidal components in the FES2012 finite-element tide model (Carrère et al., 2012) and the bottom topographic slopes instead of roughness to estimate the

energy flux at the ocean bottom (Tanaka et al., 2007). The simulated vertical diffusivities are large over rough bottom topographies and in areas with large tidal motions (Fig. 2a). The diffusivities exponentially decay in an upward direction (e.g., along 10°N in Fig. 2b). These are similar to the simulated diffusivities of Figs. 1 and 2 in the study by St. Laurent et al. (2002). The diffusivities do not change much over time because the tidal flow used to estimate the energy flux is assumed to be constant, and therefore, the diffusivities change in time only through changes in the local stratification.

We used the 3-hourly atmospheric surface dataset JRA55-do ver.08 (Tsujino et al., 2018) to estimate surface fluxes in OFES2. This dataset is based on the JRA55 atmospheric reanalysis at a horizontal resolution of approximately 55 km (Kobayashi et al., 2015). Momentum and heat fluxes are calculated with the bulk formulas proposed by Large and Yeager (2004). Note that we used the relative wind speed considering the surface current to estimate the surface momentum flux. We also included the effects of river runoff at river mouths as additional freshwater flux using a daily mean climatological

river runoff dataset from Coordinated Ocean-Ice Reference Experiments (CORE) version 2 (Large and Yeager, 2004). The





sea surface salinity (SSS) is restored to monthly climatological values of the WOA13 v2 observations (Zweng et al., 2013) with a 15-day timescale to avoid unrealistic salinity fields.

Since the polar regions are not simulated, the temperature and salinity are restored at all depths to the monthly climatological values from the same WOA13 v2 observations (Locarnini et al., 2013; Zweng et al., 2013) within a distance
of 3° from the northern and southern boundaries of the model domain. Additionally, the temperature and salinity near the straits of Gibraltar, Hormuz, and Bab el-Mandeb are restored to observations at all depths since the horizontal resolution of the model is inadequate to capture dynamics within these straits (Fig. 3). The strait of Gibraltar is where the Atlantic Ocean connects to the Mediterranean Sea and the straits of Hormuz and Bab el-Mandeb are where the Persian Gulf and the Red Sea are connected to the Indian Ocean, respectively. The water mass exchanges between these marginal seas and open oceans are
known to be important for simulating the intermediate water mass in the Atlantic Ocean and the Indian Ocean.

OFES (Sasaki et al., 2008), which follows a 50-year climatological simulation, has been integrated from 1950 to the present (Masumoto et al., 2004). OFES2 was integrated from 1958 to 2016 and started with the temperature and salinity fields of OFES from January 1, 1958. Table 1 is the list of the updates for OFES2 compared to OFES. The maximum depth of OFES2 is increased to 7,500 m from 6,065 m. The surface fluxes are now based on 3-hourly data rather than daily data to
capture the diurnal cycle. Momentum fluxes are based on a bulk formula rather than that estimated in the reanalysis. A tidal mixing scheme and a sea-ice model are newly included. River runoff is also added as additional freshwater flux. SSS is restored with a 15-day time scale rather than a 6-day timescale. The timescale was relaxed compared to OFES, where neither sea ice nor river runoff was used.

## 3  Mean oceanic fields

We next discuss improvements in the mean oceanic fields in OFES2 from OFES by comparing those to the observations. The mean temperature and salinity fields at a horizontal resolution of 0.25° averaged over 2005-2012 from the World Ocean Atlas 2013 version 2 (WOA13, Locarnini et al., 2013, Zweng et al., 2013) are used, which include a large number of Argo float observations. During this period, both OFES2 and OFES were well spun up. Satellite-observed SSH over 1993-2016 from AVISO is used to examine the simulated oceanic circulations and SSH variations in both OFES2 and OFES. To see
how the sea-ice model works in OFES2, the climatological data of sea-ice cover averaged over 2005-2012 from HadISST version 1 (Rayner et al., 2003) is compared with the data in OFES2.

### 3.1 Global oceanic fields

### 3.1.1 Sea surface temperature and salinity

Figures 4a and 4c show the 8-year mean SST and SSS biases averaged over 2005-2012 in OFES2 against WOA13. For SST
(Fig. 4a), the bias is less than 1°C in most parts of the globe. A weak cold bias broadly spreads over the subtropical Pacific and Indian Oceans and the Arctic Ocean and weak warm bias spreads over the sub-Arctic Pacific, the sub-Arctic Atlantic,





and the Southern Oceans. However, we found prominent biases in several regions. Warm biases (> 1 °C) appear in the South Pacific (170°-130° W and 55° S) and to the north of the Kuroshio Extension (140°-170° E and 35°-40° N). In the North Atlantic, along the Gulf Stream and the North Atlantic Current and in the Labrador and Norwegian Seas, several large warm
and cold biases (magnitudes larger than 1°C) are present. One possible cause of these biases is the unrealistic current pathway of the Gulf Stream. The Gulf Stream does not turn to the north at approximately 40° N, which we will examine more in detail in the next section.

The mean SSS biases in OFES2 (Fig. 4c) are smaller than 0.2 psu in most regions. This feature is partly due to the restoring boundary condition, but several large biases (larger than 0.2 psu) exist sporadically. The salty bias (> 0.4 psu) in
the North Atlantic (30° W and 50° N) likely comes from the unrealistic Gulf Stream pathway, similar to the SST bias mentioned above. The salty bias (> 0.4 psu) also appears to the north of South America and in the northern part of the Bay of Bengal, probably due to the underestimation of freshwater from the Amazon and Ganges-Brahmaputra Rivers, respectively. The river runoff in CORE version 2 (Large and Yeager, 2004) is weak, and so a more realistic product, such as JRA55-do (Suzuki et al., 2018), may be required to improve the biases. There are large fresh biases in the Arctic Ocean and large salty
bias in the Nordic Sea. Observation errors under sea ice may be partially responsible for these SSS biases at high latitudes.

Figures 4b and 4d show the 8-year mean SST and SSS biases averaged over 2005-2012 in OFES against WOA13. The SST biases are much smaller in OFES2 (Fig. 4a) than in OFES (Fig. 4b). Cold (warm) SST biases with large amplitudes appear in the equatorial and subtropical regions (high-latitude regions) in both hemispheres in OFES (Fig. 4b). The centers of the cold biases (< -1°C) zonally spread along 15° N and 15° S in the Pacific Ocean and the northwestern and southeastern
Indian Ocean. Patches of warm biases (> 1°C) exist in the Antarctic Ocean to the south of the Antarctic Circumpolar Current (ACC). Prominent warm biases (> 1°C) appear in the northwestern Pacific, the Sea of Okhotsk, and along the west coasts of South America and Southern Africa. The prominent warm biases along the west coasts in OFES are presumably associated with unrealistic coastal currents and upwelling, which are driven by unrealistic wind stresses near the coasts in the NCEP reanalysis (Fig. S1). These reductions in OFES2 are likely a result of using the bulk formula (Large and Yeager, 2004) and
the atmospheric surface data (JRA55-do), which are optimized to drive OGCMs (Tsujino et al., 2018). Additionally, the implementation of a sea-ice model in OFES2 may contribute to the decreases in the warm biases in the Arctic Ocean and the Sea of Okhotsk.

The mean SSS biases in OFES2 (Fig. 4c) are also much reduced compared to those in OFES (Fig. 4d), especially in the tropical and subtropical regions. We notice that the global distribution of the biases in OFES (Fig. 4d) is quite similar to the
difference between WOA98 (Conkright et al., 1998) and WOA13 averaged over 2005-2012 (Fig. 4f). This similarity suggests that the SSS fields in OFES2 are too much restored to WOA98. In contrast, the global distribution of the SSS biases in OFES2 (Fig. 4c) does not resemble the difference between long-term mean WOA13 and WOA13 over 2005-2012 (Fig. 4e). The weak restoring in OFES2 does not greatly constrain the simulated SSS. Therefore, the SSS bias in OFES2 (Fig. 4c) comes from something other than the restoring.



### 3.1.2 Sea surface height and its variability

Figure 5 shows the average and standard deviation of the sea surface height (SSH) over 1993-2016 in OFES2, OFES, and AVISO. The large-scale distribution of the mean SSHs in OFES2 (Fig. 5a) agrees well with that in AVISO (Fig. 5c), suggesting that OFES2 reproduces the global ocean circulations. The SSH variability (Fig. 5d) is large around the Gulf Stream, the Kuroshio, and the ACC, which also resembles that in AVISO (Fig. 5f). This large variability is mostly due to high activities of mesoscale eddies and shifts in frontal positions (e.g., Chelton et al., 2007).

However, there are regional differences in the mean SSH distribution and its standard deviation in OFES2 compared to those in AVISO. The mean SSH contours along the Gulf Stream extend towards the northeast (Fig. 5a), while a sharp northern turn is observed at approximately 40° W (Fig. 5c). The SSH variability is large along this simulated Gulf Stream (Fig. 5d). The zonal extension of the mean SSH contours along the Azores Current at approximately 33° N in the northeastern Atlantic (Fig. 5c) and large SSH variability accompanied by this current (Fig. 5f) are recognizable in AVISO but do not appear in OFES2 (Fig. 5a and Fig. 5d). For the Kuroshio in OFES2, the SSH variability is too large along the southern coast of Japan. This large variability is due to the unrealistic detachment of the Kuroshio from Kyushu. Around subtropical countercurrents in the North Pacific and the South Indian Ocean and in most regions away from the strong currents, the SSH variability is slightly smaller in OFES2 than in AVISO. We discuss these issues in Section 5.

Compared to OFES (Fig. 5b), the mean SSH in OFES2 (Fig. 5a) shows improvements. In the northern and southern subtropical gyres of the Pacific, the SSH contours are oriented more in the north-south direction in OFES (Fig. 5b) than in OFES2 and AVISO (Figs. 5a and 5c). In contrast, the subtropical gyres of the Atlantic and Indian Oceans do not have this difference. The SSH variability around strong currents such as the Gulf Stream, the Kuroshio, and the ACC (Fig. 5d) in OFES2 is comparable to AVISO observations (Fig. 5f), which reduces somewhat compared to OFES (Fig. 5e). The northwestward extent with SSH variability to the west of South Africa that corresponds to propagations of Agulhas Rings is improved in OFES2, which is too distinct in OFES due to unrealistic long-lived rings. One possible cause for the improvement in the SSH field in OFES2 is the replacement of atmospheric wind driving OFES2 by JRA55-do.

### 3.1.2 Impact of tidal mixing on water mass property

Tidal mixing is considered to mix the ocean, especially above rough-bottom topography. Previous studies have suggested that the Indonesian seas are regions where such mixing significantly impacts the water mass properties (e.g., Ffield and Gordon, 1996). Koch-Larrouy et al. (2007) demonstrated how inclusion of the local tidal mixing scheme can improve the subsurface water mass in the Indonesian seas and the eastern Indian Ocean. As mentioned in the introduction, unrealistic water mass properties in the subsurface of Indonesian Seas were one of the major biases recognized in OFES (Masumoto et al., 2008), which motivated us to add the tidal mixing scheme in OFES2.

We found that the use of the tidal mixing scheme improves the subsurface salinity biases averaged over 2005-2012 in the Indonesian seas compared to those in WOA13. At a depth of 135 m (Fig. 6a), salty bias still exists over the Indonesian seas



and to the south and southwest of the Sunda Islands, but the magnitude of this bias is less than 0.2 psu. At a depth of 325 m (Fig. 6d), the fresher bias (< -0.2 psu) spreads from the western Pacific into the Celebes Sea, gradually weakens southward, and disappears at the Makassar Strait. The saltier bias (> 0.3 psu) spreads to the south of Java and the Sunda Islands, which

decreases to approximately 0.2 psu along approximately 12° S. The cause of these biases may be partially due to the lack of nonlocal tidal mixing (e.g., Nagai et al., 2017), as discussed in a study by Sasaki et al. (2018). The impact of nonlocal tidal mixing is an interesting subject for a future study.

A comparison of subsurface salinity biases in the Indonesian seas shows significant improvement in OFES2 (Figs. 6a and 6d) from OFES (Figs. 6b and 6e). The saltier bias at a depth of 135 m is large (> 0.5 psu) in the northern Banda Sea in OFES

but is greatly reduced in OFES2. To the south of the Sunda Islands, the saltier biases are prominent both at depths of 135 m (> 0.2 psu) and 325 m (> 0.5 psu) in OFES but are greatly reduced in OFES2. This result supports the importance of tidal mixing on the water mass transformation in the Indonesian seas.

The Kuril Strait between the North Pacific and the Sea of Okhotsk is another location where previous studies (e.g., Nakamura et al., 2006) suggested the importance of tidal mixing on the water mass properties of the North Pacific

Intermediate Water (NPIW). The vertical section of salinity along 165° E in WOA13 shows that this subsurface low-salinity water is accompanied by the NPIW, which both OFES2 and OFES demonstrate well (Fig. S2). The result suggests that the tidal mixing scheme in OFES2 does not affect the water properties much, which supports the results using an eddy-permitting model by Tanaka et al. (2010). Nakamura et al. (2006) may have added too large a vertical diffusivity (2e-2 $m^2 s^{-1}$) in the strait at all depths in their OGCM, which probably led to too strong impacts on the water properties.

### 3.1 Impacts of salty outflows from marginal seas

OFES could not accurately simulate high-salinity outflows from the Mediterranean Sea, the Persian Gulf, and the Red Sea to the open ocean. To represent the impact of these outflows in OFES2, we restored temperature and salinity near the straits. Proper representations of these outflows are considered important for simulating not only the subsurface but also the surface properties (e.g., Jia et al., 2000; Prasad et al. 2001; Sofianos and Johns, 2002).

Vertical sections of salinity averaged over 2005-2012 (Fig. 7) exhibit the salty outflows at the subsurface in the Arabian Sea and the Atlantic Ocean. For the Arabian Sea, the basic influence of the outflow appears to be captured in OFES2. The longitudinal section of mean salinity crossing the mouth of the Red Sea shows that OFES2 (Fig. 7a) mostly reproduces the eastward extent of salty water (> 35.5 psu) from 46° E at approximately 700 m depth, observed in WOA13 (Fig. 7c). This feature corresponds to the salty outflow from the Red Sea. The eastward extension (> 35.5 psu), however, reaches too far at

70° E, and its depth of 700 m is too stable over the basin compared to that in WOA13. OFES2 (Fig. 7d) also generally demonstrates the southward spreading of salty outflow from the Persian Gulf: salty water (> 35.5 psu) spreads southward from 25° N above 1000 m in WOA13 (Fig. 6f). However, the high salinity core (> 35.5 psu) at a depth of 800 m is slightly too distinct and deep in OFES2 (Fig. 7d).





However, we found that OFES2 does not reproduce the salty outflow from the Mediterranean Sea into the Atlantic Ocean

well by restoring temperature and salinity near the Strait of Gibraltar. A zonal vertical section of salinity along 36° N in the eastern Atlantic Ocean in WOA13 (Fig. 7i) exhibits the westward extension of salty water (> 35.8 psu) to 25° W at approximately 1100 m depth and a thick layer with almost constant salinity of 35.7 psu over 500-1100 m depths to the west of 26° W. However, the westward extent of high salinity is still weak in OFES2 (Fig. 7g). This high salinity (> 36.0 psu) remains to the east of 9° W at depths over 1000-1500 m, where OFES2 restores salinity to the observation (Fig. 3). Why the

salty water does not spread westward much in OFES2 is unclear, but this phenomenon is possibly be connected to the bias found in the mid-ocean surface circulation in the North Atlantic (Figs. 5a and 5c). Entrainment of surface water to the Mediterranean outflow near the Strait of Gibraltar is suggested as the mechanism driving the Azores Current (Jia et al. 2000; Kida et al., 2008) and turning northward of the Gulf Stream (Jia et al., 2000).

The implementation of restoring temperature and salinity conditions at the straits resulted in significant improvements in

the Arabian Sea from OFES without the setup. OFES2 reproduces the salty outflow from the Red Sea well (Fig. 7a) but OFES does not: there is no salty water at the subsurface along 13° N in the Arabian Sea (Fig. 7b). OFES2 also greatly improved the salty outflow from the Persian Gulf (Fig. 7d) from OFES (Fig. 7e). The meridional section along 65° E shows that the salinity of subsurface salty outflow in OFES (Fig. 7e) is much fresher by 0.3-0.5 psu than that in WOA13 (Fig. 7f), and its depth of 1000 m is deeper than that in WOA13 (800 m). For the Mediterranean outflow, the improvement in OFES2

from OFES is marginal. Both OFES2 (Fig. 7g) and OFES (Fig. 7h) cannot reproduce the westward extent of the salty outflow from the Strait of Gibraltar found in WOA13 (Fig. 7i).

### 3.4 Subsurface field in the subtropical North Pacific

The subsurface water properties are sensitive to the wind stress product used. Kutsuwada et al. (2019) showed that wind stress products affect the simulated oceanic fields in an OGCM not only at the surface but also in the subsurface. In the

subtropical Western Pacific, they found that the use of QuikSCAT wind stress (Kutsuwada, 1998) in another version of OFES, called OFES QSCAT (Sasaki et al., 2006), improves the subsurface water properties compared to OFES using wind stress from NCEP reanalysis (Kalnay et al., 1996).

The vertical profile of the mean temperatures in the subtropical Western Pacific in OFES2 (red curve) mostly overlaps with that in WOA13 (black curve) (Fig. 8a). The maximum difference occurs at 280 m and is less than 1 °C. This region is

characterized by subsurface salinity maximum (e.g., Nakano et al., 2005). Its depth agrees between OFES2 and WOA13 (Fig. 8b), and its peak salinity value differs by 0.2 psu.

We found that the temperature and salinity biases in OFES2 improved significantly from OFES. In the thermocline between 50 m and 350 m depths, the temperature is much lower in OFES (Fig. 8a, blue curve) than in WOA13 (black curve). The maximum difference is approximately 6 °C at a depth of approximately 150 m. The depth of the salinity maximum is

much shallower in OFES (approximately 100 m depth) than in WOA13 (approximately 140 m depth) (Fig. 8b). The maximum difference in salinity between OFES and WOA13 is large (~0.4 psu). These biases are very similar to those found



by Kutsuwada et al. (2019) in their comparison between OFES QSCAT and OFES (their Fig. 5). As Kutsuwada et al. (2019) suggested, these large biases in OFES possibly come from the surface momentum flux. The wind stress curls from NCEP driving OFES along 10° N (blue curve in Fig. 8c) are relatively strong and fluctuate considerably in the longitudinal direction. The momentum fluxes in OFES2 (red curve in Fig. 8c) estimated by using 10-m wind in JRA55-do are comparable in amplitudes and variations to the satellite observations (red curve in Fig. 3c of Kutsuwada et al., 2019). The similarity between momentum fluxes in OFES2 and the satellite observations comes from modifications of 10-m wind in JRA55-do using the satellite observations (Tsujino et al., 2018).

**3.5 Sea-ice distribution in OFES2**

We implemented a sea-ice model in OFES2, which is not present in OFES. The domain of OFES2 excludes a large, central part of the Arctic Sea and the southern-most parts of the Ross Sea and the Weddell Sea. Figure 9 shows the distribution of monthly climatological sea-ice cover in the polar regions averaged over 2005-2012 compared to the observations from HadISST. The sea-ice cover around Antarctica in March is realistic in OFES2 (Fig. 9a). The simulated sea ice covers most areas of the Weddell Sea, as found in HadISST (Fig. 9b). A bit of sea ice remains along the most coastline of East Antarctica (right side of the figure) in HadISST. However, OFES2 misses the observed sea-ice cover near the coast from 90° E to 180° E. The sea-ice distribution in September expands greatly compared to March in HadISST (Fig. 9d), and OFES2 exhibits its distributions very well (Fig. 9c). Off the coast of Victoria Land between 180° E and 150° E (lower side of the figure) and along the southern boundary of the model domain (76° S) in the Ross Sea (160° E-150° W), the sea-ice concentration in OFES2 is somewhat lower than in HadISST.

The observed sea ice in the Arctic region in March covers the Chukchi Sea and seeps into the Bering Sea through the Bering Strait (Fig. 9f). OFES2 reproduces the observations well (Fig. 9e). However, the simulated sea ice spreads too much southward into marginal seas: the Baltic Sea, the Gulf of Saint Lawrence, and the Sea of Okhotsk. In September, unrealistic sea-ice cover spreads in the Chukchi Sea (Fig. 9g), which does not exist in HaISST (Fig. 9h).

**4 Interannual variations**

**4.1 Niño3.4 and Indian Ocean Dipole Mode indexes**

We examine the monthly time series of indexes for El Niño and IOD events to determine how well OFES2 reproduces these variations over 1968-2016 (Fig. 10 and Table 2). HadISST version 1 (Rayner et al., 2003) is used as the reference because it covers the whole analysis period. In HadISST, however, the anomalous SST in the eastern pole during the IOD events, which is discussed in Section 4.2, appears to be obscure.

The variations in the Niño3.4 index are very similar between OFES2 and HadISST (Fig. 10a). The correlation of indexes between OFES2 and HadISST is very high (0.963), and its RMS amplitude of OFES2 (0.95 °C) is slightly larger than the observed value (0.89 °C). For IOD, the Dipole Mode Index (DMI) time series in OFES2 is also similar to that in HadISST





(Fig. 10b). The correlation between the DMI of OFES2 and HadISST is high (0.714), but its RMS amplitude in OFES2 (0.52 °C) is considerably larger than that in HadISST (0.32 °C).

In OFES, the indexes of El Niño and IOD events are also similar to those in HadISST (see Table 2 for the correlations and RMS amplitudes), with somewhat lower correlations than OFES2. One possible cause for these high correlations in OFES2 is the replacement of atmospheric dataset by JRA55-do to estimate surface fluxes. The RMS amplitudes in OFES (0.93 °C for Niño3.4 index and 0.38 °C for DMI) are comparable to those of HadISST. The reason why the DMI RMS amplitude is larger in OFES2 (0.52 °C) than in OFES or the HadISST (0.32 °C) is the SSTA variations simulated in the eastern pole of

the IOD. The SSTAs in the eastern pole in OFES2 are colder (warmer) mostly in the positive (negative) IOD years of 1982, 1983, 1994, 1997, and 2006 (1996, 1998, and 2010) than OFES and HadISST (Fig. 10c). The SSTA variations show that its amplitude in OFES2 (0.43 °C) is much larger than in OFES (0.33 °C) and HadISST (0.33 °C). On the other hand, OFES2 exhibits well the time series of observed SSTA in the western pole (Fig. 10d), with the correlation of the SSTA between OFES2 and HadISST being 0.847. This value is higher than 0.751 from OFES. In OFES, the warm biases increase greatly

after 2005. The RMS amplitudes in OFES2 (0.31 °C) and HadISST (0.33 °C) are relatively small compared to that in OFES (0.41 °C). Therefore, we will closely examine this SST distribution around the eastern pole in the typical positive and negative IOD years next.

### 4.2 Sea surface temperature around the eastern pole of the Indian Ocean Dipole

We examine strong positive and negative IOD events in 1997 and 2010 as typical cases. The satellite observations captured a

low SST (< 26 °C) in the nearshore area to the southwest of Sumatra and Java during the positive event (Fig. 11c). The anomalous southeasterly wind induces coastal upwelling, and therefore, the SST near the coast becomes low compared with the offshore SST. OFES2 (Fig. 11a) reproduces this observed anomalous cold SST along the coast well, although the SST near Java is too cold (< 22 °C). During the negative event, the satellite-observed SST was warm (~30 °C) to the west of Sumatra (Fig. 11g). OFES2 (Fig. 11e) also exhibits this observed warm SST well. This warming is presumably due to weak

upwelling from the weak wind west of Sumatra (Fig. 11e). OFES2 reproduces the cold and warm SST anomalies well in the eastern pole of these and other (not shown) IOD events.

      We used the DMI estimated from HadISST version 1 (Rayner et al. 2003) as the observational index because HadISST covers a long period: 1871 to the present. However, HadISST in Fig. 11d (Fig. 11h) is incapable of capturing the cold (warm) SST near the southwestern coast of Sumatra and Java in the typical positive (negative) IOD event. Therefore, the

DMI amplitude from HadISST is likely to be too small. In contrast, OISST v2 (Reynolds 1988), covering the relatively short period from 1981 to the present, exhibits well the anomalous SST near the coast both in the positive and negative IOD events (Fig. S3), which is similar to the satellite observations (Figs. 11c and 11g). The average amplitude of the DMI over 1981-2016 is 0.54 °C for OISST v2, which is comparable to 0.54 °C for OFES2. These results suggest that OFES2 reproduces the anomalous SST near the southwestern coast of Sumatra and Java during IOD events and exhibits both the variations in and

the amplitudes of the DMI well.



OFES (Fig. 11b) did not accurately reproduce the observed anomalous cold SST (Fig. 11c) near the Sumatra and Java during the mature positive IOD event in 1997. The SST in OFES remains unrealistically warm (> 26 °C) to the southwest of Sumatra and Java. We attribute this fault to the wind stress driving OFES. The strong southeasterly wind stress (thick arrows, > 0.05 N m$^{-2}$) is located far offshore (Fig. 11b), which cannot induce coastal upwelling with realistic strength. On the other

hand, the anomalous warm SST in the eastern pole during the negative IOD in 2010 is fairly realistic in OFES (Fig. 11f), although the SST in the entire region is somewhat colder than from the satellite observations (Fig. 11g). This cold SST bias seems consistent with the bias all over the Indian Ocean in the long-term mean in OFES (Fig. 4b). The difference in the SST reproducibility in the eastern pole between the positive and negative events in OFES probably comes from the asymmetric property of the IOD events (e.g., Hong et al., 2008).

**5 Summary and discussion**

This paper describes a new version of our OGCM, which we call OFES2. OFES2 improves the atmospheric forcing to include the diurnal cycle and now includes a tidal mixing scheme and a sea-ice model. We have presented how well OFES2 simulates the mean oceanic features and interannual variations such as El Niño and IOD events, which are generally improved compared to OFES (Table 3).

OFES2 reproduces large-scale circulations, global distributions of mesoscale eddy activities, SSTs, and SSSs well with significant improvements found in the water properties in the subsurface in the subtropical Western Pacific and the Arabian and Indonesian Seas over OFES. OFES2 also represents the large SSHA RMS accompanied by strong currents well, such as the Gulf Stream and the Kuroshio, which is too large in OFES. However, the RMS values are slightly smaller in most regions in OFES2 than in the satellite observations. The surface momentum fluxes in OFES2 are estimated with a bulk

formula by using the surface wind relative to the simulated surface current. This method weakens mesoscale eddies, as Zhai and Greatbatch (2007) suggested, which possibly damps the SSHA RMS a bit too much in OFES2. Considering the atmospheric response to the SST distributions, such vertical mixing (e.g., Wallace et al., 1989) and pressure adjustment over the SST fronts (e.g., Lindzen and Nigam, 1987) in OGCM may be one of the solutions to overcome this issue.

The variations of the climate indexes of Niño3.4 and DMI are also well simulated in OFES2. The correlations of the

monthly indexes between OFES2 and the observations are slightly higher than for OFES. During the typical positive IOD event, anomalous southeast wind near the Sumatra and Java induces anomalous cold SST fields via coastal upwelling. OFES2 reproduces this anomalous SST distribution well during typical events, which is due to the realistic surface winds of JRA55-do driving OFES2.

There are several issues in OFES2 that remain unrealistic from OFES. For example, parts of the pathways of the Kuroshio

and Gulf Stream are unrealistic, which created a strong SST bias (Fig. 4a and 4b) and unrealistic SSHA RMS variability (Fig. 5d and 5e) around these currents. OFES2 uses the wind velocity relative to the surface current to estimate the surface momentum fluxes and a deeper maximum bottom depth (7,500 m), as Tsujino et al. (2013) and Kurogi et al. (2016) did to





solve these issues for the Kuroshio. Nevertheless, the simulated Kuroshio in OFES2 is frequently unrealistically detached from Kyushu. The Azores Current was also not simulated even with a restoring condition to reproduce the impact of the

salty Mediterranean overflow, which would establish the Azores Current as suggested in Jia et al. (2000). An interesting result is that the Azores Current does exist during the initial two decades (until 1970), but the current disappears after 1970. We have not yet found the cause behind such behavior nor a way to overcome this issue.

The latest supercomputer systems enable us to perform global eddying ocean simulations with much less computational cost than before. Sensitivity experiments are becoming more feasible. Sasaki et al. (2018) showed that the inclusion of a tidal

mixing scheme can result in an enhancement in ITF transport due to the basin-scale SSH increase in the tropical Pacific Ocean. While the direct impact of tidal mixing is local, its impact appears to spread over a whole basin via Rossby and Kelvin waves (Furue et al. 2015). Ensemble simulations are another way of utilizing computational power. Nonaka et al. (2016) conducted a 3-member ensemble simulation using OFES, and they suggested the existence of intrinsic variations in the midlatitude ocean currents. One future direction of global, multidecadal, eddying ocean simulation is to obtain a large

ensemble.

Global or basin-scale simulations capable of resolving oceanic submesoscales with a finer horizontal resolution (e.g., Sasaki et al. 2014, Qiu et al. 2018) are also being pursued. However, it is still difficult to execute these simulations over many decades due to the huge demands on computational resources and storage. The causes behind model biases in eddying simulations are still unresolved, and we still have much to learn from these simulations. Our improved hindcast simulation

will be useful for exploring oceanic processes and for Lagrangian analyses of water mass properties (e.g., Kida et al., 2019). We hope that OFES2 will serve as a valuable tool for studying various oceanic features with wide spatiotemporal scales from mesoscale to large-scale circulation and from intraseasonal to decadal timescales.

**Author contribution**

H. Sasaki and S. Kida implemented a tidal scheme and N. Komori implemented a sea-ice model into OFES2. H. Sasaki, S.

Kida, and R. Furue wrote the manuscript. H. Aiki, Y. Masumoto, T. Miyama, M. Nonaka, Y. Sasai, and B.Taguchi contributed model configurations and writing the manuscript.

**Code and data availability**

OFES and OFES2 are based on MOM3, which is available through https://github.com/mom-ocean/MOM3. The code has been modified for large-scale high performance simulation and implementations of sea-ice model and tidal mixing scheme.

The modification is copyrighted by Japan Agency for Marine-Earth Science and Technology (JAMSTEC). The modified code, scripts, and input data to run OFES and OFES2 are available under a copyright agreement. Monthly fields from



OFES2 become available in the immediate future. Monthly fields from OFES can be downloaded from https://doi.org/10.17596/0002029.

We thank Hiroyuki Tsujino for providing us with the earlier version of the JRA55-do dataset before the official release of
the latest version (https://esgf-node.llnl.gov/search/input4mips/). The river runoff dataset from CORE version 2 was downloaded from https://data1.gfdl.noaa.gov/nomads/forms/core/COREv2.html. The ocean bathymetry from ETOPO1 (doi:10.7289/V5C8276M) was used. WOA13 and WOA98 are available at https://www.nodc.noaa.gov/OC5/woa13/ and https://www.esrl.noaa.gov/psd/data/gridded/data.nodc.woa98.html, respectively. The HadISST was downloaded from https://www.metoffice.gov.uk/hadobs/hadisst/. AMSR-E SST version 7 and AVHRR SST version 4.1 were used through
APDRC (http://apdrc.soest.hawaii.edu/index.php). The AMSR data are produced by Remote Sensing Systems and were sponsored by the NASA AMSR-E Science Team and the NASA Earth Science MEaSUREs Program. AVHRR Pathfinder SSTs were made by GHRSST and the US National Oceanographic Data Center. The AVISO SSH data and FES2012 tidal current speeds were downloaded through Aviso (ftp.access.aviso.altimetry.fr). The DMIs for HadISST were downloaded from http://www.jamstec.go.jp/aplinfo/sintexf/iod/dipole_mode_index.html.

## Acknowledgements

OFES and OFES2 were conducted by using the Earth Simulator under the support of JAMSTEC. This work was supported by a Grant-in-Aid for Scientific Research on Innovative Areas 6102 (KAKENHI Grant No. JP19H05701) from JSPS of Japan. H. Sasaki and Y. Sasai were supported by JSPS KAKENHI (grant number 17K05662). S. Kida was supported by JSPS KAKENHI (grant number 18H03731). M. Nonaka was supported by JSPS KAKENHI (grant number 17K05665). We
appreciate Takeshi Doi, who provided information about observational data to examine IOD events.

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



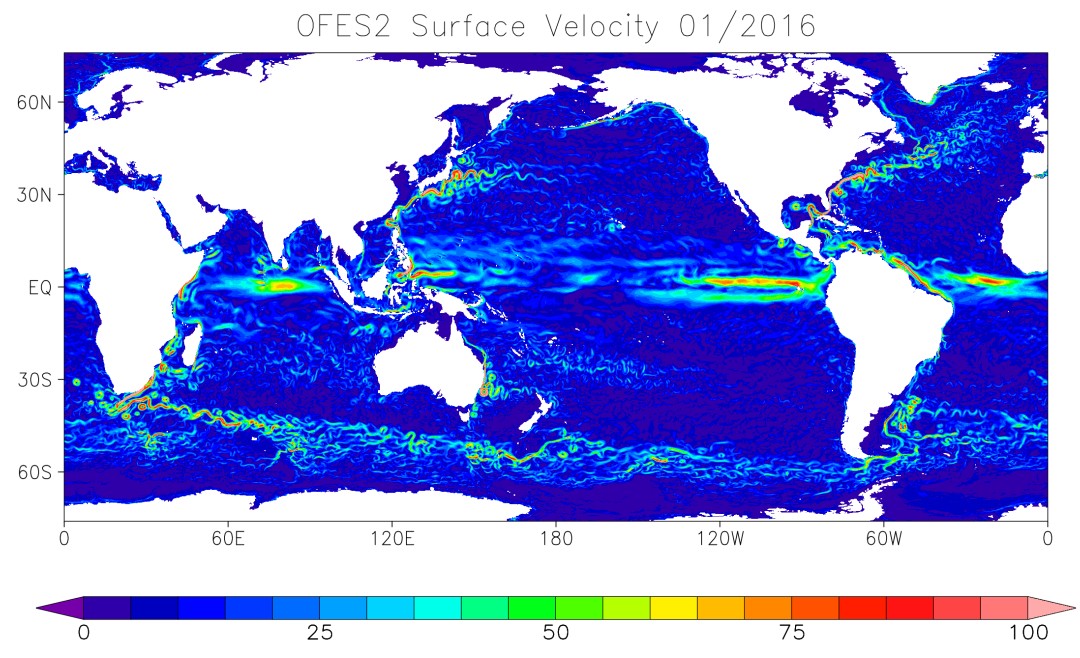


**Figure 1: An example of monthly averaged surface current speeds (cm s⁻¹) in OFES2.**

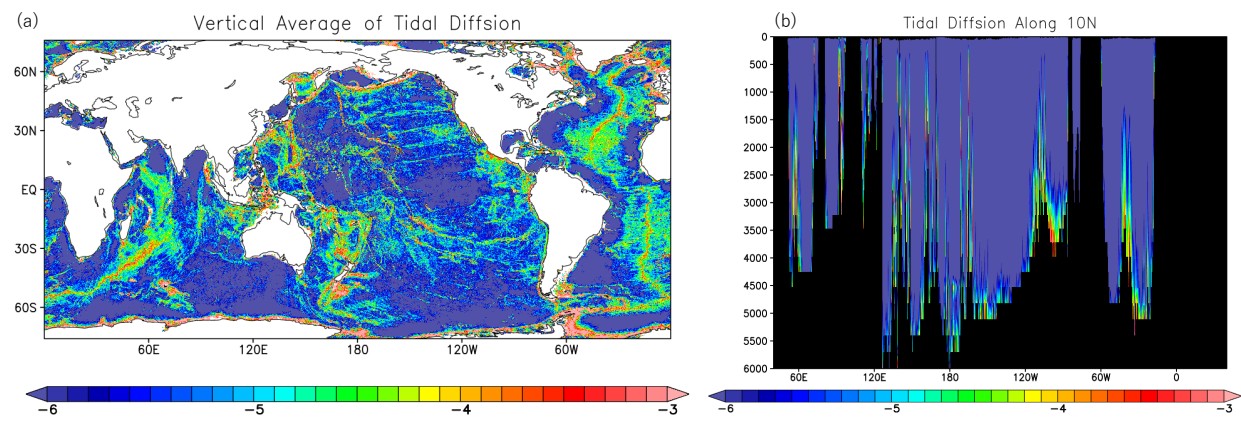

**Figure 2: Daily mean vertical diffusivity (log10 m² s⁻¹) on December 1, 2016 estimated by the tidal mixing scheme (a) vertically**

**averaged from the surface to the bottom and (b) in the vertical section along 10° N.**



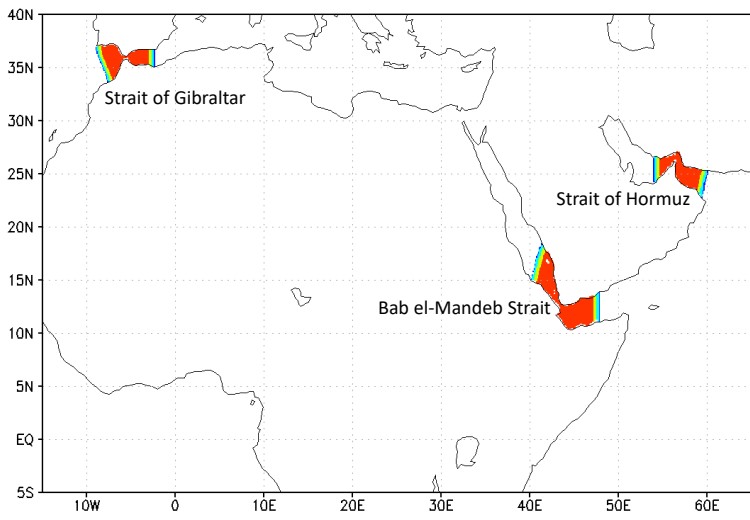

**Figure 3: Timescales for restoring the temperature and salinity in and near the Straits of Gibraltar, Hormuz, and Bab el-Mandeb. Red, yellow, light blue, and blue represent timescales of 1, 5, 10, and 30 days, respectively.**



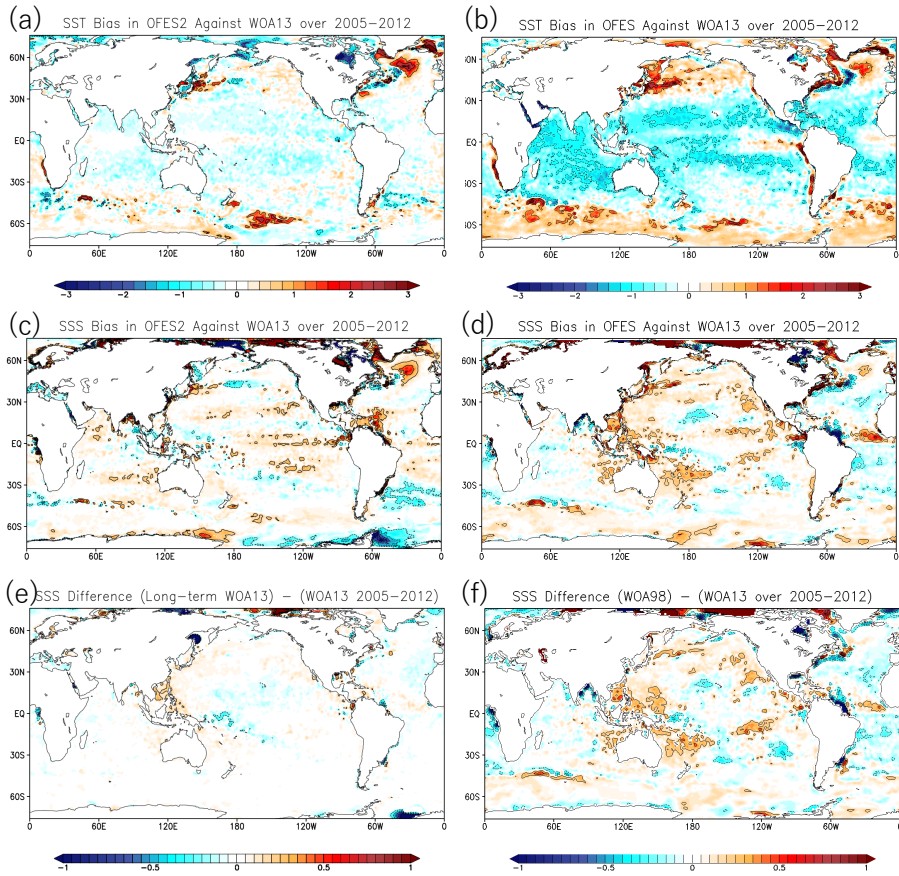

**Figure 4: SST bias (°C) in (a) OFES2 and (b) OFES averaged over 2005-2012 against WOA13. (c) and (d) are the same as in (a) and (b), respectively, but the SSS bias is shown instead (psu). SSS differences in (e) long-term WOA13 and (f) WOA98 from WOA13 averaged over 2005-2012. The contour lines are superimposed at an interval of 1 °C for SST and 0.2 psu for SSS, but zero contour lines are omitted.**

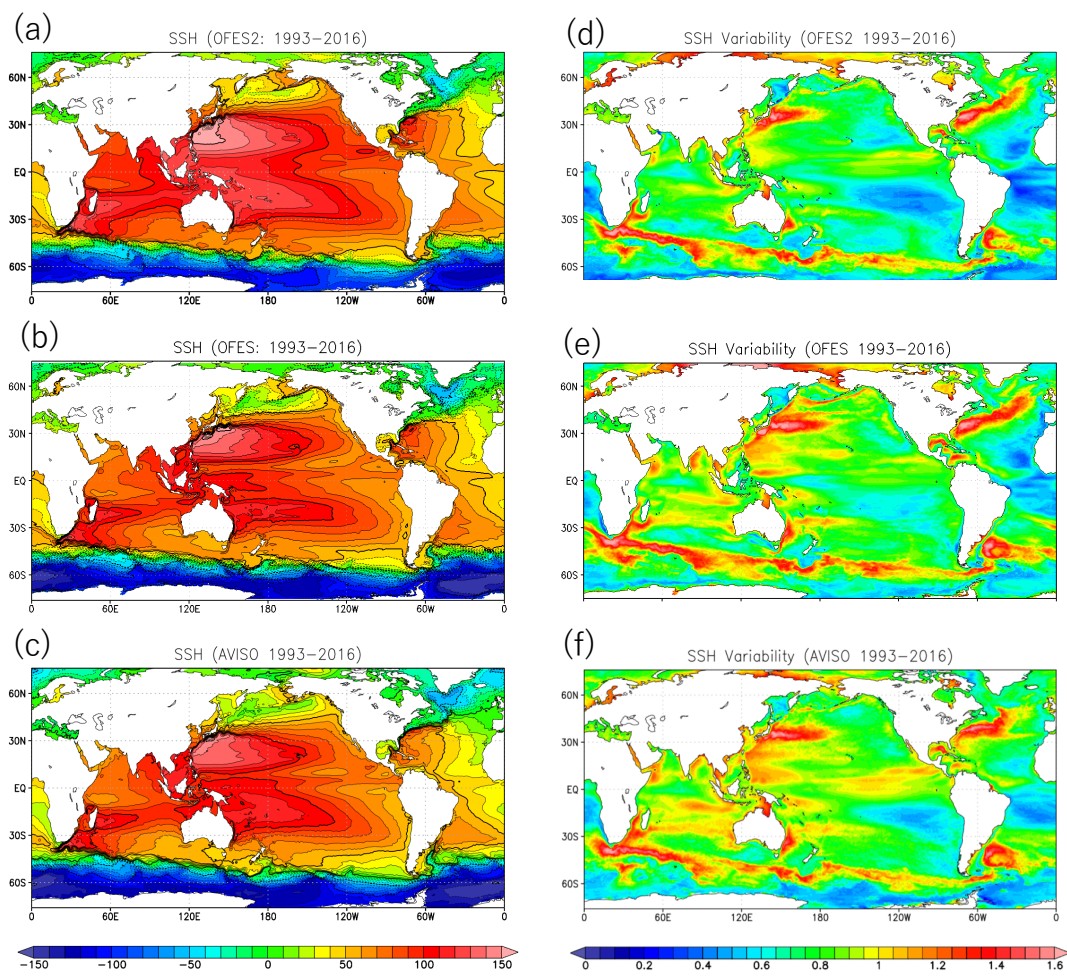

**Figure 5: Mean (a, b, c) SSH (cm) and (d, e, f) its standard deviation (log10 cm) averaged over 1993-2016 from (a, d) OFES2, (b, e) OFES, and (c, f) AVISO observations.**



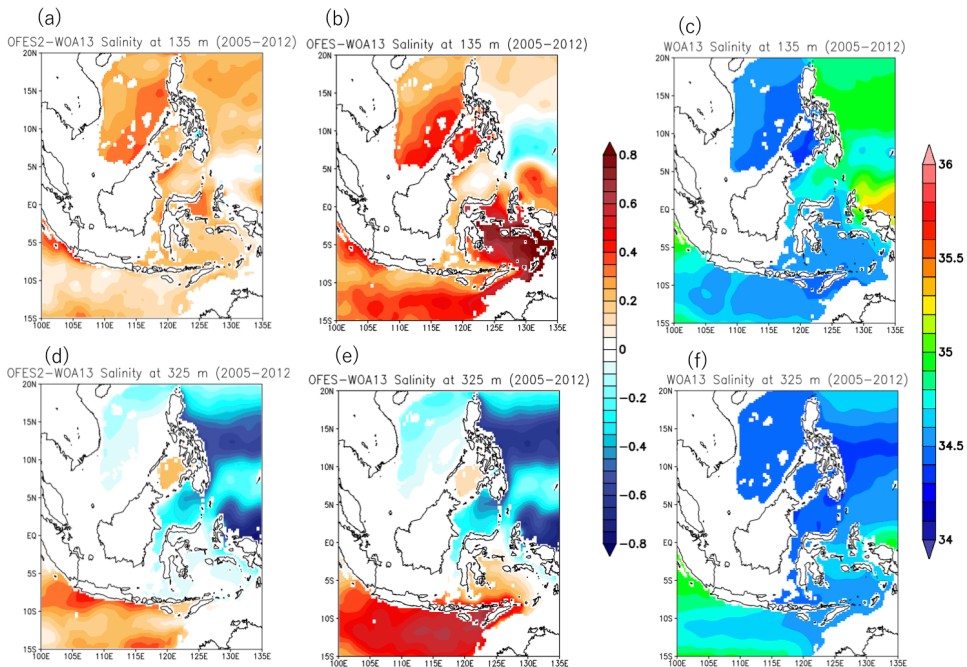

**Figure 6: Salinity biases (a, b, d, e) against WOA13 (c, f) in OFES2 (a, d), OFES (b, e) at 135 m (a, b, c) and at 325 m (d, e, f). All fields are averaged over 2005-2012, and the units are psu.**






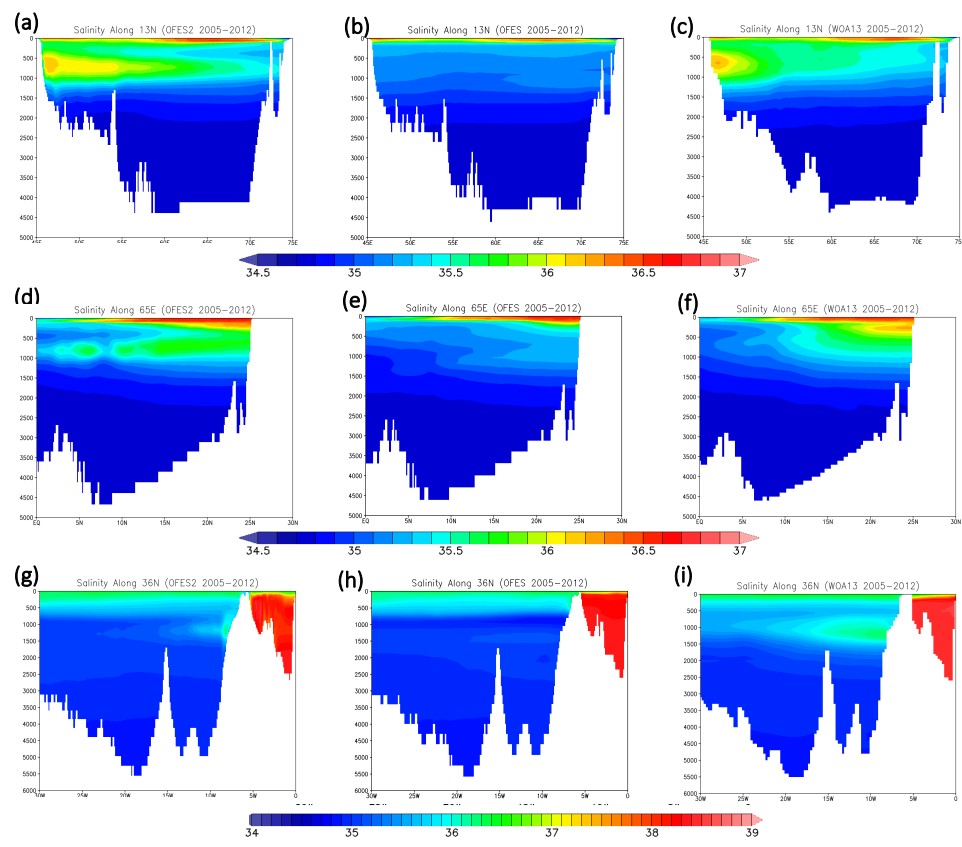

**Figure 7: Vertical sections of mean salinity along (a-c) 13° N and (d-f) 65° E in the Arabian Sea and (g-i) 36° N in the eastern Atlantic Ocean averaged over 2005-2012: (a, d, g) OFES2, (b, e, h) OFES, and (c, f, i) WOA13**

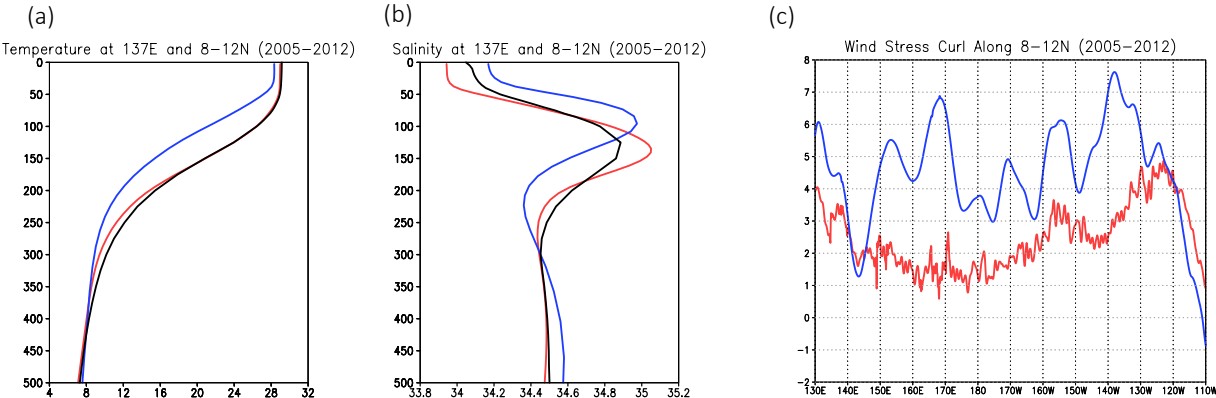

**Figure 8: Vertical profile of (a) temperature (°C) and (b) salinity (psu) at 137°E averaged from 8°N to 12°N and over 2005–2012.**
**(c) Longitudinal distributions of the wind stress curl ($10^{-8}$ N m$^{-3}$) along 10° N (averaged from 8° N to 12° N and over 2005-2012). The red, blue, and black curves are OFES2 driven by JRA55-do, OFES driven by NCEP reanalysis, and the WOA13 observations, respectively.**





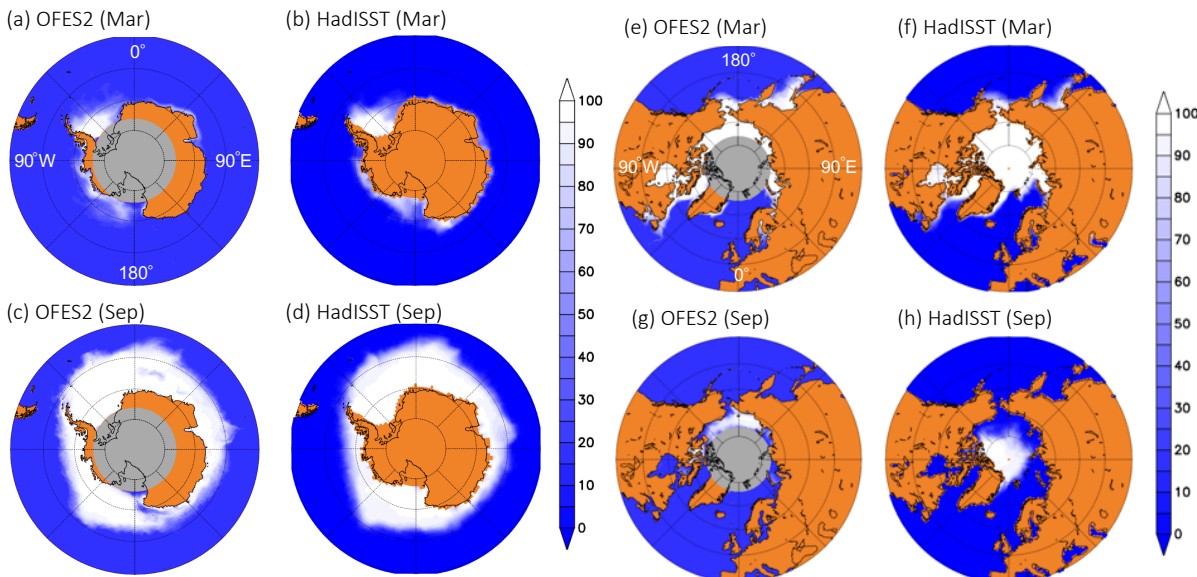

**Figure 9: Sea-ice concentrations (%) in the Antarctic Ocean in (a, b) March and (c, d) September in (a, c) OFES2 and (b, d) HadISST averaged over 2005-2012. Similarly, the sea-ice concentrations in Arctic Ocean in (e, f) March and (g, h) September in (e, g) OFES2 and (f, h) HadISST. The gray areas are out of the model domain in OFES2 (a, c, e, g).**



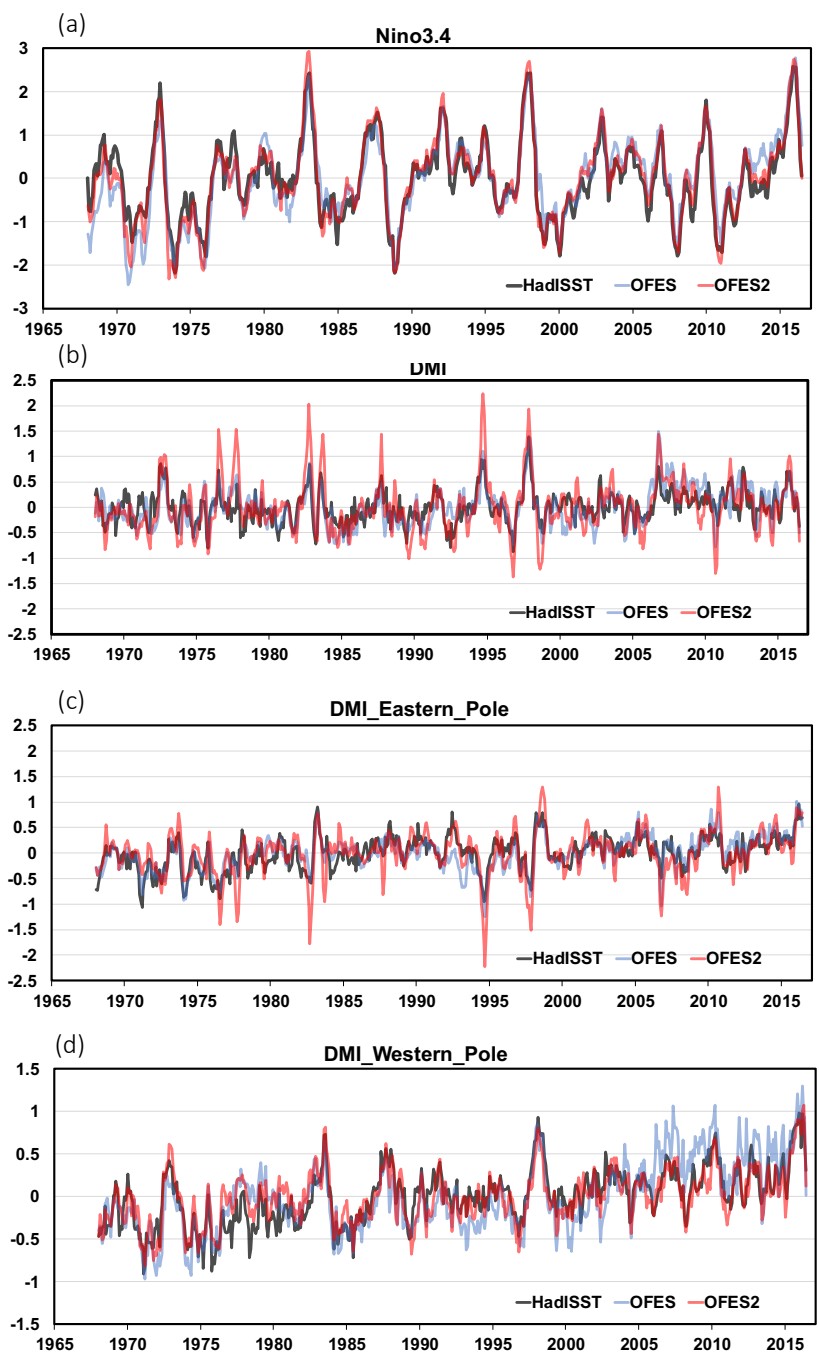

**Figure 10: (a) Monthly Niño3.4 index defined as SSTAs (°C) in 165°-145° W and 5° S-5° N in the eastern topical Pacific and (b) the monthly DMI (°C) defined as difference between the SSTAs (°C) in the (c) eastern (90° E-110° E, 10° S-0°) and (d) the western (50°-70° E and 10° S-10° N) poles (Saji et al. 1999) from OFES2 (red curve), OFES (blue curve), and HadISST version 1 (http://www.cpc.ncep.noaa.gov/data/indices/ and http://www.jamstec.go.jp/frcgc/research/d1/iod/iod/dipole_mode_index.html).**





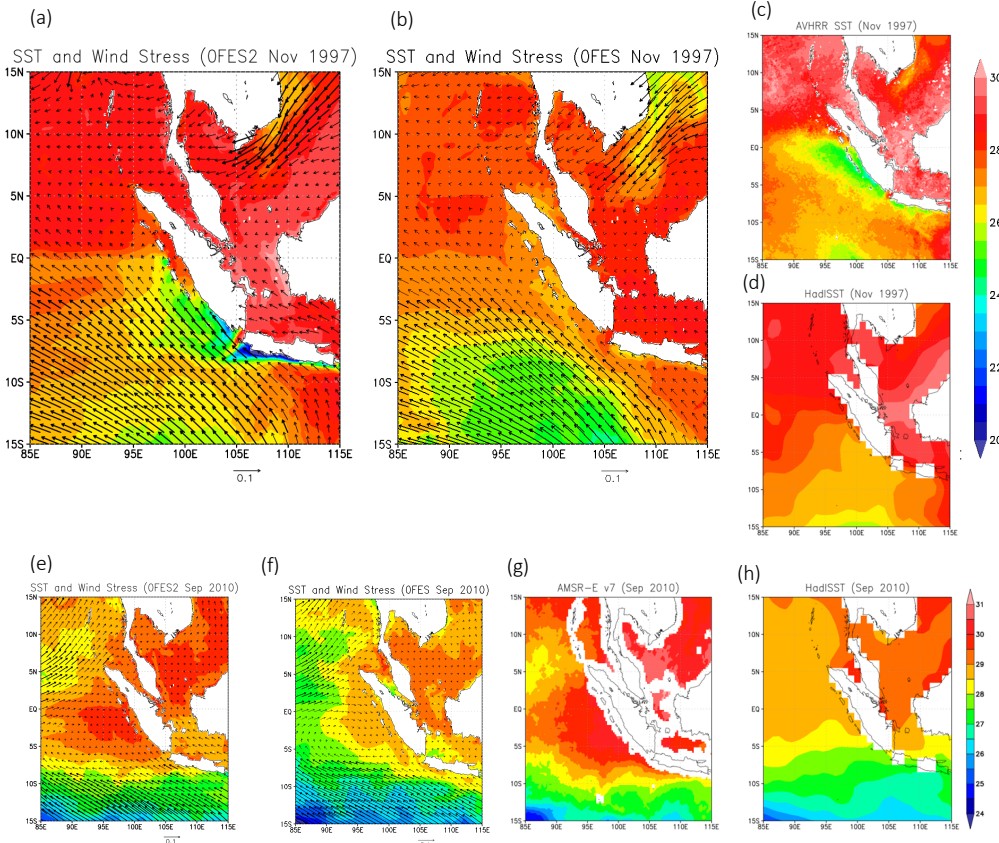

**Figure 11: SST (°C) in a region including the IOD eastern pole (90°-110° E and 10° S-0°) in the mature month of (a-d) the 1997 positive IOD event (November 1997) and (e-h) the 2010 negative IOD event (September 2010). (a, e) OFES2, (b, f) OFES, (c, g) satellite observations of AVHRR version 4.1 (Casey et al. 2010) and AMSR-E version 7 (Wentz & Meissner 2007), and (d, h) HadISST ver.1 (Rayner et al. 2003). The vectors in (a, b, e, f) are the surface wind stress (N m-2) in the models, which are plotted at a 1°×1° resolution. The thick vectors denote wind stress magnitudes stronger than 0.05 N m⁻².**



|  | OFES2 | OFES |
|---|---|---|
| Domain | 76°S-76°N | 75°S-75°N |
| Horizontal Resolution | 0.1° | 0.1° |
| Number of Vertical Levels | 105 | 54 |
| Maximum Depth | 7,500 m | 6,065 m |
| Bathymetry Data | ETOPO1 | OCCAM 30' |
| Sea-Ice Model | Komori et al. 2005 | - |
| Horizontal Mixing Scheme | Biharmonic | Biharmonic |
| Vertical Mixing Scheme | Noh & Kim 1999 | KPP (Large et al. 1994) |
| Tidal Mixing Scheme | St. Laurent et al. 2002 | - |
| SSS Restoring | 15 days to WOA13 | 6 days to WOA98 |
| Northern/Southern Artifical Boundary | T & S restoring within 3° from the boundary | T & S restoring within 3° from the boundary |
| Important Narrow Channels | Straits of Gibraltar, Hormuz, and Bab el-Mandeb | - |
| Atmospheric Forcing | JRA55-do (3 hourly, 2.5° x 2.5°) | NCEP (daily, 55km x 55km) |
| River Runoff | CORE2 (daily climatology) | - |
| Bulk Formula | Large & Yeager 2004 | Rosati & Miyakoda 1988 |
| Momentum Flux | Bulk formula using the relative wind speed | Momentum flux in NCEP (daily) |
| Hindcast Period | 1958-2016 | 1950-2017 |
| Initial Condition | T & S of OFES on Jan 1, 1958 | OFES climatlogical run |
| Outputs | Daily mean every 3 days until 1989<br>Daily mean from 1990<br>Monthly mean | Snapshot every 3 days from 1980<br>Monthly Mean |

**Table 1: Descriptions of the quasi-global eddying hindcast simulations of OFES2 and OFES.**






(a)

|  | OFES2 | OFES | HadISST |
|---|---|---|---|
| RMS Amplitude of the Niño 3.4 index(℃) | 0.95 | 0.93 | 0.89 |
| Correlation with the Niño 3.4 in HadISST | 0.963 | 0.880 | - |
| RMS Amplitude of the DMI(℃) | 0.52 | 0.38 | 0.32 |
| Correlation with the DMI in HadISST | 0.714 | 0.659 | - |

(b)

|  | OFES2 | OFES | HadISST |
|---|---|---|---|
| RMS Amplitude of the Eastern Pole DMI(℃) | 0.43 | 0.33 | 0.33 |
| Correlation with the Eastern Pole DMI in HadISST | 0.713 | 0.749 | - |
| RMS Amplitude of the Western Pole DMI (℃) | 0.31 | 0.41 | 0.33 |
| Correlation with the Western Pole DMI in HadISST | 0.847 | 0.751 | - |

**Table 2: (a) RMS amplitude (°C) of the Niño3.4 index and the DMI for OFES2, OFES, and HadISST and their correlations between OFES2 and HadISST and between OFES and HadISST. (b) Same as (a) but the eastern and western pole DMIs.**


| | Improvements in OFES2 over OFES | New or remaining issues in OFES2 |
|---|---|---|
| **SST (3.1.1)** | Suppressed cold biases in the equatorial and subtropical regions<br><br>Suppressed warm biases in the high-latitude regions, the Arctic Ocean, the Sea of Okhotsk, and along the west coasts of South America and South Africa | Warm biases in the South Pacific and to the north of the Kuroshio Extension<br><br>Warm and cold biases along the Gulf Stream |
| **SSS (3.1.1)** | Suppressed large biases by relatively weak SSS restoring | Salty biases in the North Atlantic and northern part of the Bay of Bengal and to the north of the South America |
| **Mean SSH (3.1.2)** | More realistic gyres in the subtropical North and South Pacific<br>Suppressed too distinct propagations of the Agulhas Rings | Unrealistic pathways of the Gulf Stream and Kuroshio<br><br>No Azores Current |
| **SSH variability (3.1.2)** | Suppressed too large variability along the strong currents | Slightly small in the regions away from the strong currents |
| **Water property (3.2, 3.3, 3.4)** | Suppressed biases in the subsurfaces of the Indonesian Seas, the Arabian Sea (salty outflows from the Persian Gulf and Red Sea), and the subtropical western Pacific | Lack of nonlocal tidal mixing in the Indonesian Seas<br><br>Unrealistic subsurface in the northeastern subtropical Atlantic Ocean (salty outflow from the Mediterranean Sea) |
| **El Niño and IOD (4)** | Slightly higher correlations of the indexes with observations<br>More realistic SST near the Sumatra and Java during the IOD events | |

**Table 3: Improvements in OFES2 over OFES and new or remaining issues in OFES2.**