# Peer review of "A global eddying hindcast ocean simulation with OFES2"

_Geoscientific Model Development, 2019_

## Referee Comment (RC1) · Anonymous Referee #1 · 3 Mar 2020

This paper presents description and assessment for a new quasi-global eddying ocean hindcast simulation, OFES2, conducted by a Japanese ocean modeling group. Compared to the previous version (OFES), OFES2 has incorporated several physical processes such as a sea-ice model and a tidal mixing scheme into the model, and is forced by a new surface atmospheric dataset. As a result, many improvements are identified in OFES2 relative to OFES, motivating further analyses using OFES2.

It is very important for users of the OFES2 dataset if a reference paper describing the model settings and simulation results is provided. Thus, I am highly supportive of publication of this paper.

In the following, I will raise several specific points that I would like to ask authors to consider before the paper is published.

[Figure]

L.94 and Table 1 (River runoff): If I recall correctly, COREv2 dataset is providing monthly mean climatology of river runoff.

Table 1 (Atmospheric Forcing): Please swap horizontal resolutions between JRA55-do and NCEP.

L.112: It would be informative if the restoring time scales for SSS are normalized as days over a 50 m length scale and compared with Table 2 of Danabasoglu et al. (2014).

L.156: "OFES2" should perhaps read "OFES".

L.161 (Figure 5): Please describe the standardization, or specifically global offsetting, applied to the model and AVISO sea levels. Is a common area used to compute offsetting factors for models and AVISO?

L.182: I think that the "eddy killing" effect of using relative wind for computing wind stress (e.g., Renault et al. 2019) is contributing to the improved representation of the northward extent of SSH variability west of South Africa.

L.183: The section number should perhaps read "3.2", not "3.1.2".

L.207: My take on Figure S2 is that NPIW in OFES2 is slightly improved relative to OFES presumably owing to incorporating the tidal mixing scheme.

L.210: The section number should perhaps read "3.3", not "3.1".

L.230: Then, what kind of processes or parameterizations are needed to represent the entrainment of surface water to the Mediterranean outflow near the Strait of Gibraltar in models?

L.342: Please insert "as" after "such".

L.343: Recently, Renault et al. (2018) showed that the surface oceanic currents make imprints on the surface atmospheric winds through surface momentum flux in the cou-

pled atmosphere-ocean system. Accordingly, it would not necessarily be appropriate to subtract the full surface oceanic current from the surface wind vector in computing surface momentum fluxes in uncoupled ocean simulations. Several recipes are compared and discussed by Renault et al. (2020). If the approach of correcting relative wind is used (Note that this approach is not the most recommended recipe by Renault et al. 2020), a value around 0.7 should be multiplied to the surface ocean current before it is subtracted from the surface wind for computing surface momentum fluxes. It would be appropriate to refer to these recent works here. Also, it might be of interest for both authors and users to conduct a sensitivity experiment applying this modification to the method of computing momentum fluxes and to check its impact on SSH variability in OFES2 in the future.

L.346: "southeast wind" should perhaps read "southeasterly wind".

L.355-357: I thought that it would not be rather difficult for the authors to check and report in this paper whether the realistic water mass structure present in the initial condition is gradually decaying by 1970 or the applied forcing is destroying the structure abruptly around 1970.

References:

- Danabasoglu et al. (2014): Ocean Modelling, https://dx.doi.org/10.1016/j.ocemod.2013.10.005

- Renault et al. (2018): Scientific Reports, https://doi.org/10.1038/s41598-017-17939-1

- Renault et al. (2019): Geophysical Research Letters, https://doi.org/10.1029/2018GL081211

- Renault et al. (2020): Journal of Advances in Modeling Earth Systems, https://doi.org/10.1029/2019MS001715

---

## Referee Comment (RC2) · Anonymous Referee #2 · 3 Mar 2020

This paper presents a newly released OFES2 ocean model and some improved model results compared with the old OFES. A sea-ice model and a tidal mixing scheme are included with the new JRA55-do atmospheric forcing. Sea surface temperature and sea surface salinity are both improved in many key regions. Specifically, the new atmospheric forcing improves the eastern Indian Ocean sea surface temperature. However, the authors attribute to the change of atmospheric data in many improvements without further convincing evidences. Also, this manuscript should provide a more general overview of this product (e.g., Pacific/Atlantic/Indian Ocean) to the readers instead of many regional discussions. In general, this manuscript will be used as the future reference for OFES2 and is appropriate to be published in GMD after considering the following comments.

1. Introduction: p2, other global eddy-resolving models are briefly mentioned here. However, the authors should comment on the differences and the unique characteristics of the new OFES2 among them. It seems the main OFES2 is to update the old OFES.

2. Section 2 requires further detailed description for the new features because this paper will be representative for OFES2 in the future. Line 71, is the thickness within the upper 500m non-uniform? Can you describe more about the coupling of sea-ice model? Can you comment on the impact of without polar region on the sea ice model and the ocean model? Is the sea-ice model coupled to the ocean model internally or through a coupler? This is important because the model results may be very sensitive to the coupling frequency. There are many issues related to the sea-ice model. The authors have to address this clearly.

3. In addition, can you clarify if you replace the original KPP by the Noh and Kim (1999)? Then, add the new tidal mixing scheme. If so, why do you replace the KPP? Any specific reason? For the tidal mixing, why do you include only K1 and M2? Are they the dominant components everywhere or specifically for the Indonesia region?

4. Section 2: The most important change is the inclusion of the JRA55-do. The big difference is the use of relative wind speed. Can the author quantify how big the final wind stress is changed to force the ocean? Also, what's the global distribution of this improvement overall? This is important because the authors contribute many improved results to this updated dataset. Also, does the double-counting of the ocean current suggested in Sun et al. (2019) exist? Is the NECC still weak in the OFES2?

Sun, Z., Liu, H., Lin, P., Tseng, Y.‐h., Small, J., & Bryan, F. (2019). The modeling of the north equatorial countercurrent in the community earth system model and its oceanic component. Journal of Advances in Modeling Earth Systems, 11, 531– 544.

5. line 98, what's the time scale used in the temperature and salinity restoring? Any sponge layer? Is this the reason causing the large SSS biases near the northern

boundary (Figures 4c and 4d)? If you impose the restoring, why do you still get such a large bias there? Should this affect the coupling with sea-ice model? These issues need to be addressed in a greater detail.

6. Line 104: do you have any results to support this? This manuscript only emphasizes on the Indonesian and Arabian Seas. Unfortunately, these are very regional validation. The watermasses in the Atlantic, Pacific and Indian Ocean should be discussed.

7. Section 3.5 It seems the Sea-ice distribution is generally reasonable. However, the interannual variation and its long-term trend are what we care most. Is the long-term trend consistent with the observation? What's the major advantage of bringing this sea-ice model if Arctic region is not simulated? Also, can the missing of North pole cause any artifacts in the northern boundary? Particularly, does this have any impact on the deep water formation? Should the reader need to be caution for any potential problem?

8. Lin128-132, it seems the eddy resolving model still cannot resolve the Kuroshio and Gulf Stream separation in OFES2. The authors need to comment on this further (e.g., Schoonover et al., 2016; 2017; McWilliams et al., 2019). Although a brief discussion here and next paragraph, I cannot see any useful information from the discussion. McWilliams, J. C., Gula, J., & Molemaker, M. J. (2019). The gulf stream north wall: Ageostrophic circulation and frontogenesis. Journal of Physical Oceanography, 49( 4), 893– 916. Schoonover, J., and Coauthors, 2016: North Atlantic barotropic vorticity balances in numerical models. J. Phys. Oceanogr., 46, 289–303 Schoonover, J., Dewar, W. K., Wienders, N., & Deremble, B. (2017). Local sensitivities of the gulf stream separation. Journal of Physical Oceanography, 47, 353– 373.

9. Line 136-137, is this really due to the inclusion of weak river runoff (underestimation)? Or is it possible due to the vertical/horizontal mixing? These regions have not only the positive biases but also negative biases if you look carefully. Particularly, negative biases are mostly along the coast. This seems not "underestimation".

[Figure]

10. Line 138, are you sure more realistic product help? OFES (no inclusion of river runoff) shows negative biases in these regions. Maybe the estuary circulation mixing or tidal mixing is more important?

11. Line 140, "Observation errors …" Can you clarify this further? Why this comes from the observation errors? The northern boundary is restored to the observation, right? This is confusing.

12. Line 150-152, is this correct? If so, why not in salinity? The atmospheric surface data change should contribute mainly to the momentum flux, which should impact both temperature and salinity, right?

13. P5, last sentence, what is this "something"? Also, the northern boundary is re-stored to the observation. Why the model results are not converging to the observation?

14. Line 182, how and why replacing to JRA55-do change the SSH directly? Is this error very common in other ocean models using the same forcing? I don't think so. The authors need to clarify this further.

15. Section 3.1.2 shows the impact of tidal mixing on water mass property. However, the author only show the results of Indonesian Seas and Eastern Indian Ocean. What about other key regions? Most important of all, can this deteriorate other regions?

16. P7, first 2 paragraph, I suggest to reduce these paragraphs or provide some more new information. The message for these two paragraphs is the salty biases are re-duced in the subsurface in OFES2. That's it.

17. P8, what's the main purpose of using the restoring in these marginal seas? The niche is the restoring can help regionally if the process is local (like Persian Gulf and Red Sea). However, it is well-known that the Mediterranean overflow can affect the Atlantic Overturn Circulation while the restoring cannot capture its overflow process therefore, the restoring can cannot help the simulation. It is not clear why the authors

consider this approach here. It doesn't help fundamentally.

18. Section 3.4, line 248-251. What about other subsurface regions? Why this region is chosen? Is this the region where the large difference of JRA55-do and NCEP product? I suggest the authors to show the regions of the largest and smallest differences. Also, the tropic is a well-known region that the wind correction is largest (Large and Yeager, 2004). It may be better to use these specific regions to show the impact of differences. Otherwise, it is just hand waving to say every improvement comes from the atmospheric wind changes.

19. Line 258-263, why do you think this improvement in OFES2 come from the momentum flux change? Why not other flux? Can you provide a more convincing evidence? If this is the main cause, the momentum fluxes should change both temperature and salinity, right? It seems only temperature is greatly improved but salinity is not.

20. Section 3.5, it seems the sea-ice model is also missing for the polar region. Then, . Is adding the sea-ice model affect the large-scale general circulation? Any global impact or is it just a regional impact? A key question is that does adding the sea-ice model improve the deep water formation and overall model performance?

21. Section 4.1 why are these two indices are chosen? Why not checking the AMO or other important ocean indices? Before discussing the interannual variation, I suggest the authors to discuss the long-term trend first. This is very important for the first order evaluation.

22. Line 292, again, why? Here, the authors contribute the biases to the JRA55-do without further information. Any result to support this speculation for the good flux?

23. Line 321-329, here, the author attribute the improvement to the coastal upwelling resulting from the winds. Is this a general cause? I suggest the authors to replace (or add) the particular year by the IOD composite years (i.e., positive composite years and negative composite years). This may support the discussion here.

Minor comments: 1) The labels in Section 3 look strange. 3.1(3.1.1, 3.1.2, 3.1.2) 3.1, 3.4. They are totally messed up. 2) Line 179 "which reduces" changed to "which is reduced" 3) Line 208, remove "a" after "large. Also -2 should be the superscript.
* * *

---

## Author Comment (AC1) · 26 Mar 2020

Monthly fields from OFES2 have become available through https://doi.org/10.17596/0002029. The DOI of OFES2 data is the same as that of OFES data.

---

## Author Response (AR1)

**Response to Reviewer #1's comment.**

We thank reviewer #1 for carefully reading the manuscript and for the supportive comments. We have carefully considered your comments and revised our manuscript. Our item-by-item replies follow below.

*L.94 and Table 1 (River runoff): If I recall correctly, COREv2 dataset is providing monthly mean climatology of river runoff.*

We used a monthly mean river runoff from COREv2, as you pointed out. We corrected the description of the dataset (new Line 104 in the revised manuscript with revision tracking) and Table 1.

*Table 1 (Atmospheric Forcing): Please swap horizontal resolutions between JRA55-do and NCEP.*

Thank you for letting us know the typo. We corrected the horizontal resolutions in Table 1.

*L.112: It would be informative if the restoring time scales for SSS are normalized as days over a 50 m length scale and compared with Table 2 of Danabasoglu et al. (2014).*

The SSS restoring time scales of 15-day and 6-day correspond to 150-day and 60-day with respect to 50 m depth respectively. We added these normalized time scales in new Line 121.

*L.156: "OFES2" should perhaps read "OFES".*

Thank you for letting us know the typo. We corrected it.

*L.161 (Figure 5): Please describe the standardization, or specifically global offsetting, applied to the model and AVISO sea levels. Is a common area used to compute offsetting factors for models and AVISO?*

The mean SSH in both OFES and OFES2 was offset by adding 50 cm. We added a description about this offsetting in the caption of Fig. 5.

*L.182: I think that the "eddy killing" effect of using relative wind for computing wind stress (e.g., Renault et al. 2019) is contributing to the improved representation of the northward extent of SSH variability west of South Africa.*

Thank you for letting us know a possible contribution of "eddy killing" to the improvement of SSH variability corresponding to the Agulhas Rings. We added the contribution (new Lines 198-200) with a reference to Renault et al (2019).

*L.183: The section number should perhaps read "3.2", not "3.1.2".*

Thank you for letting us know the typo. We corrected it.

*L.207: My take on Figure S2 is that NPIW in OFES2 is slightly improved relative to OFES presumably owing to incorporating the tidal mixing scheme.*

**We added the slight improvement of the NPIW in OFES2 compared to that in OFES (new Lines 226-227).**

*L.210: The section number should perhaps read "3.3", not "3.1".*

**Thank you for letting us know the typo. We corrected it.**

*L.230: Then, what kind of processes or parameterizations are needed to represent the entrainment of surface water to the Mediterranean outflow near the Strait of Gibraltar in models?*

**We are not aware of any simple solution. Note that Azores Current does exist during the initial two decades (until 1970) when the Mediterranean overflow is presented well in OFES2. These discussions were included in the discussion section (new Line 398).**

*L.342: Please insert "as" after "such".*

**Thank you for letting us know the typo. We corrected it.**

*L.343: Recently, Renault et al. (2018) showed that the surface oceanic currents make imprints on the surface atmospheric winds through surface momentum flux in the coupled atmosphere-ocean system. Accordingly, it would not necessarily be appropriate to subtract the full surface oceanic current from the surface wind vector in computing surface momentum fluxes in uncoupled ocean simulations. Several recipes are compared and discussed by Renault et al. (2020). If the approach of correcting relative wind is used (Note that this approach is not the most recommended recipe by Renault et al. 2020), a value around 0.7 should be multiplied to the surface ocean current before it is subtracted from the surface wind for computing surface momentum fluxes. It would be appropriate to refer to these recent works here. Also, it might be of interest for both authors and users to conduct a sensitivity experiment applying this modification to the method of computing momentum fluxes and to check its impact on SSH variability in OFES2 in the future.*

**Thank you for letting us know about two papers and your suggestions. We referred both studies and added such sensitivity experiments as one of our future works in the discussion section (new Lines 373-375).**

*L.346: "southeast wind" should perhaps read "southeasterly wind".*

**Thank you for letting us know the typo. We corrected it.**

*L.355-357: I thought that it would not be rather difficult for the authors to check and report in this paper whether the realistic water mass structure present in the initial condition is gradually decaying by 1970 or the applied forcing is destroying the structure abruptly around 1970.*

We plotted the SSH distribution and vertical section of salinity in the Atlantic Ocean averaged over the 1960s, 1970s, 1980s, and 1990s and added them as appendix figures (Figs. S6 and S7 in new Line 397). These figures show that the Azores Current and Mediterranean  salty outflow existed in the 1960s but the both abruptly decayed in the 1970s and disappeared after the 1980s.

 **Response to Reviewer #2's comment.**

We thank reviewer #2 for carefully reading the manuscript and providing useful comments. We have carefully considered the comments and revised our manuscript. Our item-by-item replies follow below.

*This paper presents a newly released OFES2 ocean model and some improved model results compared with the old OFES. A*
80  *sea-ice model and a tidal mixing scheme are included with the new JRA55-do atmospheric forcing. Sea surface temperature and sea surface salinity are both improved in many key regions. Specifically, the new atmospheric forcing improves the eastern Indian Ocean sea surface temperature. However, the authors attribute to the change of atmospheric data in many improvements without further convincing evidences. Also, this manuscript should provide a more general overview of this product (e.g., Pacific/Atlantic/Indian Ocean) to the readers instead of many regional discussions. In general, this manuscript*
85  *will be used as the future reference for OFES2 and is appropriate to be published in GMD after considering the following comments.*

The primary focus of OFES2 is the upper ocean circulation up to decadal timescales. Considering its integration time of ~50 years, you can see that deeper circulation shouldn't be much different from that implied from the initial condition. For this reason, we focus on SST, SSS, and SSH for the general overview of the OFES2 output (Section 3.1) before
90  moving on to the specific issues. For the latter, we choose water mass in the Indonesian Seas, the salty outflow from marginal seas, and subsurface fields in the subtropical North Pacific (Section 3.2, 3.3, and 3.4) because those were not well reproduced in the previous version of our model (Introduction, new Lines 53–58 in the revised manuscript with revision tracking) and are markedly improved in OFES2. Regarding the contributions of the atmospheric data, it is well known that in general, the upper ocean circulation is strongly constrained by surface heat flux, P−E flux, and
95  winds. Therefore, the quality of atmospheric data is a key to simulate more realistic oceanic field in the OGCM. We respond to your specific comments below.

*1. Introduction: p2, other global eddy-resolving models are briefly mentioned here. However, the authors should comment on the differences and the unique characteristics of the new OFES2 among them. It seems the main OFES2 is to update the old*
100  *OFES.*

One of the uniquenesses of OFES is that the outputs are publicly available and widely used (more than 300 papers published by using the outputs, http://www.jamstec.go.jp/res/ress/sasaki/ofes_publication.html), which was emphasized (new Lines 43-44). OFES2 is an updated hindcast simulation of the widely used OFES.

105  *2. Section 2 requires further detailed description for the new features because this paper will be representative for OFES2 in the future. Line 71, is the thickness within the upper 500m non-uniform? Can you describe more about the coupling of sea-ice model? Can you comment on the impact of without polar region on the sea ice model and the ocean model? Is the sea-ice model coupled to the ocean model internally or through a coupler? This is important because the model results may be very*

*sensitive to the coupling frequency. There are many issues related to the sea-ice model. The authors have to address this*
110  *clearly.*

**We added further detailed descriptions of the model. The thickness of each layer within the upper 100 m is 5 m. The thickness gradually increases and 55 levels exist within the upper 500 m (new Lines 74-74). The sea-ice model was internally implemented into OFES2 (new Line 77-78). OFES2 mostly covers the Antarctic Ocean but does not include most of the Arctic Ocean. SSS in the Chukchi, Nordic and Labrador Seas (new Lines 150-151) and sea-ice in the**
115  **Chukchi Sea (new Lines 301) is unrealistic, probably due to the artificial northern boundary (new Line 152-153, 301-302).**

*3. In addition, can you clarify if you replace the original KPP by the Noh and Kim (1999)? Then, add the new tidal mixing scheme. If so, why do you replace the KPP? Any specific reason? For the tidal mixing, why do you include only K1 and M2?*
120  *Are they the dominant components everywhere or specifically for the Indonesia region?*

**We updated the vertical mixing scheme by replacing the KPP (Large et al., 1994) with Noh and Kim (1999) to use a more modern mixing scheme (new Lines 118-119). We included K1 and M2 as the largest impacts of diurnal and semidiurnal tidal circulations respectively in most regions (new Lines 87-88).**

125  *4. Section 2: The most important change is the inclusion of the JRA55-do. The big difference is the use of relative wind speed. Can the author quantify how big the final wind stress is changed to force the ocean? Also, what's the global distribution of this improvement overall? This is important because the authors contribute many improved results to this updated dataset. Also, does the double-counting of the ocean current suggested in Sun et al. (2019) exist? Is the NECC still weak in the OFES2?*

**The largest difference in wind stress naturally occurs where the surface ocean current is fastest. Therefore, if we take**
130  **the Kuroshio current as an example, the relative difference will be (10 m/s - 1 m/s)^2 / (10 m/s)^2 = 80% for an atmospheric wind of 10 m/s and an ocean current of 1 m/s.  To explore the impacts of this difference would need a sensitivity study, which we cannot perform. There have, however, been several papers for the sensitivity of the ocean circulation to the use of relative winds (e.g. Renault et al. 2020). In the revised manuscript we cite these papers in the discussion section (New lines 373–375).**

135  **The NECC in OFES2 (attached figure) is simulated well compared to observations (Fig. 2 in Johnson et al. 2002), although the use of relative wind to estimate the wind stress corresponding to the double-counting of the ocean current is include in OFES2. This scheme is often adopted lately, although some problem is also pointed out (Sun et al. 2019).**

[Figure]

**Figure. Eastward velocity along (a)180° E, (b) 140° W, and (c) 110°W (right) averaged over 1985-2000 in OFES2.**

*5. line 98, what's the time scale used in the temperature and salinity restoring? Any sponge layer? Is this the reason causing the large SSS biases near the northern boundary (Figures 4c and 4d)? If you impose the restoring, why do you still get such a large bias there? Should this affect the coupling with sea-ice model? These issues need to be addressed in a greater detail.*

**We added "The restoring time-scale linearly increases from 1 day at the boundary to infinity at the inner end of the restoring band." (new Lines 105-106). One of the reasons causing the large biases near the northern boundary is due to the salinity restoring toward the long-term mean WOA13, which is different from the reference SSS of WOA13 averaged over 2005-2012. Another possible reason is the unrealistic sea-ice cover in the Chukchi Sea in September in OFES2 (Fig. 9g). It is possible that the salinity restoring affects the sea ice distribution near the boundary (new Lines 301-302).**

*6. Line 104: do you have any results to support this? This manuscript only emphasizes on the Indonesian and Arabian Seas. Unfortunately, these are very regional validation. The watermasses in the Atlantic, Pacific and Indian Ocean should be discussed.*

**We deleted the sentence because this sentence is not needed in the model description section.**

*7. Section 3.5 It seems the Sea-ice distribution is generally reasonable. However, the interannual variation and its long-term trend are what we care most. Is the long-term trend consistent with the observation? What's the major advantage of bringing this sea-ice model if Arctic region is not simulated? Also, can the missing of North pole cause any artifacts in the northern boundary? Particularly, does this have any impact on the deep water formation? Should the reader need to be caution for any potential problem?*

**The reason for the sea ice model implementation is that we tried to simulate the oceanic fields in the Antarctic Ocean and the Sea of Okhotsk more realistically. We regard the sea ice in the Arctic Ocean as much better than nothing. We plotted the sea ice concentrations in 1980s (new supplemental figure Fig. S3). In the observations, the sea ice in the Arctic Sea decreased much in September in 2005-2012 compared with in 1980s. OFES2 could not capture this definitive**

**long-term trend probably because the domain of OFES does not include most of the arctic ocean. Possible artifacts near the northern boundary are the SSS bias (new Lines 150-151) and unrealistic sea-ice in the Chukchi Sea (new Lines 300-301). In the Antarctic Ocean, the long-term trend of the sea ice is much less than that in the Arctic Ocean in either OFES2 and observations (new Lines 305-306).**

170    **The primary focus of OFES2 is on the upper ocean circulation with variability on intra-seasonal to decadal time scales, rather than the meridional overturning circulation. Nonetheless, we examined the impact on the deep water formation and plotted the global and Atlantic meridional overturning (supplemental Fig. S8). Model results look reasonable (new Lines 460-462).**

175    *8. Lin128-132, it seems the eddy resolving model still cannot resolve the Kuroshio and Gulf Stream separation in OFES2. The authors need to comment on this further (e.g., Schoonover et al., 2016; 2017; McWilliams et al., 2019). Although a brief discussion here and next paragraph, I cannot see any useful information from the discussion.*

   **We added the discussions about the unrealistic Gulf Stream in OFES2 considering the Schoonover et al. 2016, 2017 and McWilliams et al. 2019. We also added Chassignet and Xu (2017), who succeeded in simulating the Gulf Stream**

180    **separation in HYCOM at a horizontal resolution of 1/50 degree. To solve this issue, sensitivity experiments as in the previous studies would be needed. We added these discussions (new Lines 389-393).**

   *9. Line 136-137, is this really due to the inclusion of weak river runoff (underestimation)? Or is it possible due to the vertical/horizontal mixing? These regions have not only the positive biases but also negative biases if you look carefully.*

185    *Particularly, negative biases are mostly along the coast. This seems not "underestimation".*

   **Thank you very much for your careful looking. We agree that these biases may be due to inaccurate mixing or coastal circulation (new Lines 149-150).**

   *10. Line 138, are you sure more realistic product help? OFES (no inclusion of river runoff) shows negative biases in these*

190    *regions. Maybe the estuary circulation mixing or tidal mixing is more important?*

   **We deleted the sentence.**

   *11. Line 140, "Observation errors : : :" Can you clarify this further? Why this comes from the observation errors? The northern boundary is restored to the observation, right? This is confusing.*

195    **We deleted the sentence.**

*12. Line 150-152, is this correct? If so, why not in salinity? The atmospheric surface data change should contribute mainly to the momentum flux, which should impact both temperature and salinity, right?*

**The contribution of atmospheric data change to SST bias improvement is correct because the SST in the OGCM is generally strongly constrained by atmosphere dataset via the surface flux. It is also true that momentum flux influences SST and SSS. We therefore added the discussion of the possible contributions of the atmospheric data and bulk formula used in OFES2 to the SSS bias reductions in the next paragraph (new Lines 168-169).**

*13. P5, last sentence, what is this "something"? Also, the northern boundary is restored to the observation. Why the model results are not converging to the observation?*

**We now provide specific examples of processes other than restoring, such as the unrealistic pathways of the Kuroshio and Gulf Stream and the unrealistic sea-ice distribution in the Chukchi Sea (new Lines 174-175). The SSS is not much constrained by the observation. Because the SSS restoring is weak, which is incorporated only to avoid long-term salinity drift. And the SSS near the northern boundary is not also much constrained by the observation because the strong restoring is limited to the area very near the boundary.**

*14. Line 182, how and why replacing to JRA55-do change the SSH directly? Is this error very common in other ocean models using the same forcing? I don't think so. The authors need to clarify this further.*

**Line 182 is incorrect as you suggested. Instead, we propose a reason of the improvement of SSHA variations along the Agulhas Ring propagation. The relative wind is used to estimate the momentum flux in OFES2, which tend to damp eddies as Renault et al. (2017, 2019a) suggested. This eddy killing effect improved the SSHA variations (new Lines 198-200).**

*15. Section 3.1.2 shows the impact of tidal mixing on water mass property. However, the author only show the results of Indonesian Seas and Eastern Indian Ocean. What about other key regions? Most important of all, can this deteriorate other regions?*

**One of the focus of this study is to examine the improvement in the large salinity bias in the Indonesian Seas and Eastern Indian Ocean found in the previous version OFES, as mentioned in the introduction. We added a comment that water mass properties in other regions need to be looked at in future studies (new Lines 403-404).**

*16. P7, first 2 paragraph, I suggest to reduce these paragraphs or provide some more new information. The message for these two paragraphs is the salty biases are reduced in the subsurface in OFES2. That's it.*

**We shortened the sentences in these two paragraphs and reduced them to one paragraph.**

*17. P8, what's the main purpose of using the restoring in these marginal seas? The niche is the restoring can help regionally if the process is local (like Persian Gulf and Red Sea). However, it is well-known that the Mediterranean overflow can affect the Atlantic Overturn Circulation while the restoring cannot capture its overflow process therefore, the restoring cannot help the simulation. It is not clear why the authors consider this approach here. It doesn't help fundamentally.*

**Our purposes of the restoring in these marginal seas is to improve the strong salinity bias found in OFES for these regions, as mentioned in the Introduction. We are aware that this approach is crude and cannot capture the detailed dynamics of overflows, but our primary goal was to avoid large bias in the temperature and salinity. We also expected that Azores Current, which was missing in OFES, will appear in OFES2 by the Mediterranean restoring, as Jia et al. (2000) successfully reproduced the Azores Current by using a North Atlantic model with similar restoring, as described in new Line 339. On the other hand, we did not focus on the potential impacts of the Mediterranean overflow on the Atlantic Meridional Overturning Circulation (Reid, 1979, McCartney and Mauitzen 2001) because our integration time of ~50 years is not sufficient to look at such impacts. We nevertheless examined the meridional overturning circulation in the Atlantic Ocean (Fig. S8) and added discussion about the issue (new Lines 399-404).**

*18. Section 3.4, line 248-251. What about other subsurface regions? Why this region is chosen? Is this the region where the large difference of JRA55-do and NCEP product? I suggest the authors to show the regions of the largest and smallest differences. Also, the tropic is a well-known region that the wind correction is largest (Large and Yeager, 2004). It may be better to use these specific regions to show the impact of differences. Otherwise, it is just hand waving to say every improvement comes from the atmospheric wind changes.*

**The subtropical Northwestern Pacific was focused following Kutsuwada et al. (2019), as mentioned in the Introduction. They found the largest temperature and salinity biases in the subsurface along 10N in OFES in their Fig. 4 (new Line 265-266) and showed the cause of these biases due to the large difference of wind stress curl in NCEP compared to QuikSCAT satellite observation. When OFES was forced with QuikSCAT, the subsurface field became much comparable to observations. OFES2 shows improved subsurface properties that resembles observations and since OFES2 is forced by JRA55-do, which has wind stress curl comparable to QuikSCAT, we consider changes in atmospheric wind as the primary cause for the improvement (Fig. 8c of Kutsuwada et al. 2019 and Fig. 3c) (new Lines 282-286).**

*19. Line 258-263, why do you think this improvement in OFES2 come from the momentum flux change? Why not other flux? Can you provide a more convincing evidence? If this is the main cause, the momentum fluxes should change both temperature and salinity, right? It seems only temperature is greatly improved but salinity is not.*

**Both subsurface temperature and salinity in OFES2 were improved as mentioned (new Line 274). In OFES, strong wind stress curl of NCEP caused strong Ekman pumping, which resulted in anomalously shallow thermocline (new**

265 **Lines 280-282). The thermocline depth is more realistically simulated in OFES2 because of the improvement in wind stress curl (as mentioned above) and therefore, temperature and salinity are both more realistic.**

*20. Section 3.5, it seems the sea-ice model is also missing for the polar region. Then,. Is adding the sea-ice model affect the large-scale general circulation? Any global impact or is it just a regional impact? A key question is that does adding the sea-*
270 *ice model improve the deep water formation and overall model performance?*

    **The impact of no sea-ice model for the polar region appears to be limited to the region. We examined the strength of the global and Atlantic meridional overturnings (Fig. S8) as a metric of the impacts of the sea-ice model on the global-scale deep circulation. The overturning streamfunctions appear realistic and similar to those in OFES (Fig. 4 in Masumoto et al. 2004). Either the impacts of sea-ice model on the basin-scale deep circulation are small or more**
275 **integration time is necessary before they show up. This discussion was included in the discussion section (new Lines 399-404).**

*21. Section 4.1 why are these two indices are chosen? Why not checking the AMO or other important ocean indices? Before discussing the interannual variation, I suggest the authors to discuss the long-term trend first. This is very important for the*
280 *first order evaluation.*

    **The primary focus of OFES2 is the upper ocean circulations with variability up to decadal variations as mentioned in the beginning. This is why we chose Nino 3.4 and IOD indices for comparison. We also examined the monthly AMO indexes (Fig.12) and found that the variations in OFES2 correlate better with observation more than those in OFES. Although the model integration of 1958-2016 in OFES2 is rather short to examine the multidecadal oscillation, we**
285 **added this interesting result in the discussion section (new Lines 380-383).**

*22. Line 292, again, why? Here, the authors contribute the biases to the JRA55-do without further information. Any result to support this speculation for the good flux?*

    **We attribute the higher correlations of Nino3.4 and IOD in OFES2 to the replacement of the atmospheric data by**
290 **JRA55-do because the simulated SST in ocean models is in general strongly constrained by the atmospheric data via the surface flux. We therefore now mention this aspect as a possible reason (new Lines 320-321).**

*23. Line 321-329, here, the author attribute the improvement to the coastal upwelling resulting from the winds. Is this a general cause? I suggest the authors to replace (or add) the particular year by the IOD composite years (i.e., positive composite years*
295 *and negative composite years). This may support the discussion here.*

    **We choose years 1997 and 2010 as representative cases for the positive and negative IOD events, but our conclusions generally apply to other positive and negative events. In the revised manuscript, we provide the composites suggested by the reviewer as a supplementary figure (S4).**

300     *Minor comments: 1) The labels in Section 3 look strange. 3.1(3.1.1, 3.1.2, 3.1.2) 3.1,*

      *3.4. They are totally messed up.*

        **We corrected the labels in Section 3.**

      *2) Line 179 "which reduces" changed to "which is reduced"*

305        **We corrected the phrase by changing to the passive form.**

      *3) Line 208, remove "a" after "large. Also -2 should be the superscript.*

        **We corrected them.**

[revised manuscript text omitted]

---

## Author Response (AR2)

**Dear Dr. Qiang Wang,**

**Thank you very much for sending us the reviews of our revised manuscript.**

*The reviewer is satisfied with your revision, but it is suggested that you need to proofread the English writting, especially for the parts added during the last revision. Although you can have further chance to proofread your manuscript after it is accepted, in this case, I would suggest you to do it first before I accept your paper. A few examples provided by the reviewer are given below.*

**We carefully checked and improved the English writing in not only the parts added in the last revision but also the other parts as much as possible. We also fixed the several typos. Please see the details in the revised manuscript with the track changes. Our modifications do not change any contents in our paper.**

*Some minor issues/typos from my quick checks.*

*Line 79-80, negative sign "-" should be "~" or just use "approximately".*

**We deleted the negative sign.**

*Line 187, "the subtropical gyres……" changed to "the improvement in the subtropical gyres of the Atlantic and Indian Oceans is limited".*

**We corrected it.**

*Line 371, should add "has" before "a deeper".*

**We corrected it.**

*Line 373, remove "frequently" and add "all the time" after Kyushu.*

**We corrected the sentence: "the simulated Kuroshio in OFES2 is frequently makes an unrealistic excursion away from Kyushu.".**

*Line 376, change "makes significant" to "contribute mainly to the".*

**We corrected it.**

*Line 378, add well before "simulated".*

**We corrected it.**

[revised manuscript text omitted]